- 1 Reconstruction of the reservoir water level-storage volume
- <sup>2</sup> relationship based on the capacity loss induced by sediment
- <sup>3</sup> accumulation and its impact on flood control operation
- <sup>4</sup> Qiumei Ma<sup>1</sup>, Chengyu Xie<sup>1</sup>, Zheng Duan<sup>2</sup>, Yanke Zhang<sup>1\*</sup>, Lihua Xiong<sup>3</sup>, Chong-Yu
- $5 Xu^4$
- <sup>6</sup> School of Water Resources and Hydropower Engineering, North China Electric Power
- University, Beijing 102206, China
- <sup>8</sup> <sup>2</sup> Department of Physical Geography and Ecosystem Science, Lund University,
- 9 Sölvegatan 12, 223 62 Lund, Sweden
- <sup>10</sup> State Key Laboratory of Water Resources Engineering and Management, Wuhan
- University, Wuhan 430072, China
- <sup>12</sup> Department of Geosciences, University of Oslo, P.O. Box 1047 Blindern, N-0316 Oslo,
- 13 Norway
- 14 Correspondence: Yanke Zhang (<u>zhangyk@ncepu.edu.cn</u>)
- 15 **Abstract.** Sediment accumulation in reservoirs can change the predefined water level—
- storage volume (WLSV) relationship by significantly reducing the storage capacity,
- which threatens the flood control safety of reservoirs in long-term scheduling and
- operation. However, reconstructing the WLSV relationship has long been challenging,
- particularly on a large scale, because traditional field bathymetric measurement is
- difficult. To fill this knowledge gap, this study proposes a method to estimate the
- reservoir WLSV curve based on the capacity loss induced by sediment accumulation.
- To assess the potential negative impact caused by an inaccurate WLSV curve, flood
- regulations are calculated for individual reservoirs using six design flood hydrographs
- with return intervals of 200–10,000 years as reservoir inflow. The flood regulation risk
- is quantified using the maximum flood regulation water level (Z\*) and the ratio of
- periods when the design flood level is exceeded ( $\gamma$ ). Based on over 10 years of
- hydrological data, sediment data, and operational information, a cascade of nine
- reservoirs in the Wujiang River in China was selected to apply the established method.

The results showed that sediment accumulation was more severe in reservoirs in the middle and upper reaches of the Wujiang River, which caused the most significant decrease in capacity loss volume for the Wujiangdu Reservoir (172.8 million  $m^3$ ) and the largest loss rate for Suofengying Reservoir (25.02%). The current design WLSV curve for flood regulation calculations underestimated  $Z^*$  by 7.11 and 1.84 m and  $\gamma$  by 2% and 3% for Suofengying and Dongfeng reservoirs, respectively, compared with the reconstructed one. This underestimation increased with the length of the return interval, which indicates that when storage capacity considerably decreases, continued use of the existing design WLSV curve may significantly underestimate the flood regulation risks and consequently pose potential safety hazards to the reservoir and downstream flood protection objects.

## 1 Introduction

In the past century, more than 50,000 large reservoirs have been constructed worldwide and regulate more than half of the global major river systems, with a total storage capacity of over 8000 billion m³ (Hanasaki et al., 2006). By operating these hydraulic structures, humans have significantly increased the volume of fresh water stored on the Earth's surface to help communities mitigate flood risks, generate clean energy, and secure a stable water supply (Castelletti et al., 2012; Şen, 2021; Zhou et al., 2021). The volume of water stored in these reservoirs exceeds 20% of the world's annual average runoff and is equivalent to three times the annual average water storage in all global river channels; thus, they are crucial elements in the hydrological and biochemical cycles (Yassin et al., 2019). The number of reservoirs under construction and planning is rapidly increasing to satisfy the demands of population growth and socioeconomic development, particularly in developing countries (Zhong et al., 2020). For example, China has the most reservoirs in the world, where over 100,000 facilities were built by 2024 (Wang et al., 2024) and the aggregate storage capacity of reservoirs in China surpassed 1060.59 billion m³ (Song et al., 2022).

Reservoir water storage is a critical water balance component and serves as

fundamental information for high-efficiency reservoir water management (Gao, 2015; Liu et al., 2022). The primary method to estimate reservoir water storage values during routine operations is to refer to the water level–storage volume (WLSV) curve, which quantifies the functional relationship between water level and storage based on gauge observations of the water level (Cao and Liu, 2018). Therefore, the accuracy and reliability of the WLSV curve are crucial in determining the characteristic storage capacities of reservoirs, such as the normal storage capacity and flood control storage capacity, which directly affect the effectiveness of subsequent reservoir operations and management decisions. The water storage capacity of a reservoir can change during long-term operation because of multiple natural and anthropogenic factors, including river diversion, sand mining, and particularly sediment accumulation (Li et al., 2011). Sediment carried by upstream inflows can be trapped by the dam and thus accumulate in the reservoir, continuously reducing its effective storage capacity. If the extent of sediment accumulation and the resulting change in storage capacity are not investigated in a timely manner, the outdated design WLSV curve will continue to be used, potentially leading to a decline in the functionality and overall benefits of the reservoir or even causing safety incidents.

Two main methods have traditionally been used to reconstruct WLSV functional relationships in previous studies. The first method involves conducting topographic surveys in the reservoir to estimate its storage capacity. For example, Sawunyama (2006) conducted field measurements of the water depth and its coordinates in the reservoir to establish a power function between surface water area and storage capacity. Zhang et al. (2011) used an in–situ bathymetric survey approach to measure the water depth of a high–altitude lake and create isobathic maps to calculate the storage capacity. However, the field measurement method is often limited by long survey durations, complex topographic conditions, and high costs, which collectively make it difficult to update the WLSV curve. In recent years, the launch of high–resolution optical and synthetic aperture radar (SAR) satellites have advanced satellite remote sensing technology and positioned it as a cutting–edge tool in hydrology. These technologies have gradually been applied to reservoir and dam monitoring, but this process remains in its

preliminary stages, and the accuracy must be improved (Gourgouletis et al., 2022). By extracting reservoir areas from remote sensing images and combining them with satellite radar altimetry, the relationships among reservoir water levels, areas, and storage volumes can be established to reconstruct the elevation—area—volume curves (Gao et al., 2012; Guan et al., 2021; Li et al., 2020; Zhang et al., 2017).

The second method is based on measured hydrological data to estimate and revise the WLSV curve using the water balance principle. Specifically, the WLSV curve can be derived from the initial storage capacity state, water level, and concurrent storage increments detected from hydrological data measured during the dry season (Fowe et al., 2015). Wang (2011) used an indirect approach based on the water balance and inflow and outflow measurements to reconstruct the WLSV curve. The selected data affect the accuracy of the WLSV curve that is reconstructed using water balance relations, and the available fluctuating range of water levels and concurrent storage volumes remains constrained because the reservoir operation rarely reaches the theoretical extreme low— and high—water level points on the WLSV curve. Due to these limitations, the WLSV curve has not been systematically rechecked in considerable reservoirs worldwide, particularly in developing countries, despite the criticality of ensuring sustainable reservoir governance.

Notably, reservoir sedimentation disrupts the natural sediment transport along river systems. As a primary driver of storage capacity reduction, inaccurate sediment estimates of reservoirs can compromise the reliability of flood control operations (Sedláček et al., 2022). Global assessments of reservoirs from recent studies show a sediment accumulation rate of 0.06–1.17% per year and a capacity loss rate of 13–19% over decades, where reservoirs in developing countries are disproportionately affected (Perera et al., 2022; Yao et al., 2023). Ran et al. (2013) and Zhang et al. (2023) similarly concluded that China lost the most reservoir storage capacity in the world with an average capacity loss rate above 11.27%, which is 2–4 times the global average. Consequently, tracking and quantifying the sediment trapped by reservoirs are critical to understanding the flood response dynamics after capacity alterations (Ren et al., 2024). Defined as the instantaneous ratio of the intercepted sediment to the total

sediment load, the trap efficiency has been a key sedimentation parameter since its conceptualization by Brown (1944). The trap efficiency was extensively investigated using multiple approaches to incorporate the reservoir capacity, watershed characteristics, and sediment load data. Then, the trap efficiency yields distinct estimation methods including the Brown, Brune, and Gill methods (Moragoda et al., 2023; Ren et al., 2024; Tan et al., 2019). However, the validity of reconstructing WLSV curves based on sediment accumulation remains understudied (Jia et al., 2021). Meanwhile, critical knowledge gaps remain regarding the differences in flood control operations when the WLSV curves significantly change. Reconstruction of the WLSV curve using storage capacity loss estimates induced by sediment accumulation provides a crucial supplement for the second traditional reconstruction method mentioned above (Huang et al., 2018).

This study aims to estimate the WLSV relationship of reservoirs by proposing a storage capacity loss rate (LR) indicator based on sediment accumulation and identify the impact of the reservoir WLSV relationship on flood control operations by

storage capacity loss rate (LR) indicator based on sediment accumulation and identify the impact of the reservoir WLSV relationship on flood control operations by quantifying the flood regulation risk. A cascade of nine reservoirs in the Wujiang River of China was selected as a case study to estimate the reservoir WLSV curve and analyze its impact. Specifically, we seek to address the following scientific questions: (a) What are the characteristics of sediment accumulation in different reservoirs in the selected cascade? (b) Can the reservoir WLSV curve be effectively reconstructed using the proposed water storage capacity LR framework based on long-term water and sediment data? (c) What is the impact of the outdated design WLSV curve on flood control operations when the reservoir storage capacity significantly decreases?

The remainder of this paper is organized as follows. Section 2 describes the cascade reservoirs in the Wujiang River Basin with the observed water and sediment data. Section 3 introduces the estimation of the LR indictor, reconstruction of the WLSV curve, and detection of flood control operation responses on the WLSV curves before and after reconstruction. Section 4 presents and discusses our analysis results. Finally, Section 5 provides the primary conclusions of this study.

## 2 Study area and data

## 2.1 Study area

Yangtze River, and it is characterized by a considerable natural elevation drop and abundant hydropower resources. The Wujiang River Basin has a catchment area of 87,920 km<sup>2</sup>, a length of 1037 km, a natural fall of 2124 m, and a subtropical wet monsoon climate. The integral role of this basin in the "West-East Electricity Transmission" project establishes its prominence as the critical basin among the 13 major hydropower bases of China (Wu et al., 2018). The cascade reservoir system of the Wujiang River in this study comprises nine reservoirs with varying shapes, surface areas, and regulation capacities. The Wujiang River exhibits relatively stable annual runoff. The Hongjiadu (HJD) Reservoir in the upstream section acts as the "leading reservoir" of the selected cascade reservoir system and offers multi-year regulation capability. The midstream Wujiangdu (WJD) Reservoir provides annual regulation, whereas the downstream Goupitan (GPT) Reservoir supports multi-year regulation. Figure 1 illustrates the spatial distribution of reservoirs in the Wujiang River Basin. After the gradual completion of the cascade system in the basin, more sediment has accumulated. As a result, during long-term operation, the storage capacity of the ninereservoir cascade system has decreased to varying degrees, which negatively affects the operation and scheduling of the reservoirs. Before 1990, the mean annual sediment accumulation in the reservoirs along the Wujiang River was approximately 9 million tons and was predominantly affected by large reservoirs; small and medium-sized reservoirs contributed a minor fraction to the total capacity. From 1991 to 2005, the primary reservoirs that intercepted sediment on the Wujiang River mainstream transitioned to the HJD and Dongfeng (DF) reservoirs, which were impounded in 2004 and 1994, respectively. The average multi-year sediment accumulation in the reservoirs on the Wujiang River was 16 million tons, which was higher than that during 1956-1990, and the proportion of sediment accumulation in major reservoirs also increased. Since 2006, although the reservoirs at

The Wujiang River is the largest tributary on the southern bank of the upper reaches of

the lower reaches of the Wujiang River have gradually been completed and begun to operate, the HJD and DF reservoirs in the upper reaches remain the primary reservoirs for sand retention (Yuan et al., 2022).

**Figure 1.** Location of the Wujiang River Basin and cascade reservoirs (HJD, DF, SFY, WJD, GPT, SL, ST, GLQ, and DHS denote the Hongjiadu, Dongfeng, Suofengying, Wujiangdu, Goupitan, Silin, Shatuo, Geliqiao and Dahuashui reservoirs, respectively. Same as follows.)

## 2.2 Data

This study collected fundamental hydrological and sediment data (such as the sand content in the inflow water, inflow, and outflow) and reservoir operation information (including the total storage capacity, design flood level, and calibration flood level) for the nine–reservoir cascade system. A professional engineering organization under a data-sharing agreement provided the water and sediment data in this study under a data-sharing agreement. We conducted rigorous quality control, including manual verification and rectification of anomalous and erroneous data. Table 1 shows the main parameters of each reservoir. The duration of the selected hydrological and sediment data was 10–23 years during 2001–2023 because different reservoirs had different

Table 1. Basic reservoir information of the Wujiang River

| Reservoir         Total storage capacity (108 m³)         Regulating capacity (108 m³)         Normal storage capacity (108 m³)         Dead storage capacity (108 m³)         Regulating capacity (108 m³)         Normal storage storage water flood flood level level level level (m)         Calibration flood level level level (m)           HJD         49.47         11.36         33.61         1140.00         1076.00         1141.34         1145.40           DF         10.25         3.74         4.91         970.00         936.00         975.69         977.53           SFY         2.01         1.01         0.67         837.00         822.00         837.97         842.37           WJD         23.00         7.80         13.60         760.00         720.00         760.30         762.80           GPT         64.54         26.62         29.02         630.00         590.00         632.89         638.36           SL         15.93         8.88         3.17         440.00         431.00         445.15         449.45           ST         9.21         4.83         2.87         365.00         353.50         366.73         369.65           DHS         2.77         1.15         1.355         868.00         845.00         868.43         871.35 <th></th> <th></th> <th></th> <th></th> <th><u> </u></th> <th></th> <th></th> <th></th>                                                     |           |                      |                      |            | <u> </u> |         |         |             |
|------------------------------------------------------------------------------------------------------------------------------------------------------------------------------------------------------------------------------------------------------------------------------------------------------------------------------------------------------------------------------------------------------------------------------------------------------------------------------------------------------------------------------------------------------------------------------------------------------------------------------------------------------------------------------------------------------------------------------------------------------------------------------------------------------------------------------------------------------------------------------------------------------------------------------------------------------------------------------------------------------------------------------------------------------------------------------------------------------------------------------------------------------------------------------------------------------------------------------------------------------------------------------------------------------------------------------------------------------------------------------------------------------------------------------------------|-----------|----------------------|----------------------|------------|----------|---------|---------|-------------|
| Reservoir         storage capacity (108 m³)         water level level level (m)         level level (m)         le | ъ :       | Total                | Dead                 | Dagulating | Normal   | Dead    | Design  | Calibration |
| capacity (108 m³)         capacity (108 m³)         level (m)                            |           | storage              | storage              |            | storage  | water   | flood   | flood level |
| HJD 49.47 11.36 33.61 1140.00 1076.00 1141.34 1145.40 DF 10.25 3.74 4.91 970.00 936.00 975.69 977.53 SFY 2.01 1.01 0.67 837.00 822.00 837.97 842.37 WJD 23.00 7.80 13.60 760.00 720.00 760.30 762.80 GPT 64.54 26.62 29.02 630.00 590.00 632.89 638.36 SL 15.93 8.88 3.17 440.00 431.00 445.15 449.45 ST 9.21 4.83 2.87 365.00 353.50 366.73 369.65 DHS 2.77 1.15 1.355 868.00 845.00 868.43 871.35                                                                                                                                                                                                                                                                                                                                                                                                                                                                                                                                                                                                                                                                                                                                                                                                                                                                                                                                                                                                                                      | Reservoir | capacity             | capacity             |            | level    | level   | level   | (m)         |
| DF       10.25       3.74       4.91       970.00       936.00       975.69       977.53         SFY       2.01       1.01       0.67       837.00       822.00       837.97       842.37         WJD       23.00       7.80       13.60       760.00       720.00       760.30       762.80         GPT       64.54       26.62       29.02       630.00       590.00       632.89       638.36         SL       15.93       8.88       3.17       440.00       431.00       445.15       449.45         ST       9.21       4.83       2.87       365.00       353.50       366.73       369.65         DHS       2.77       1.15       1.355       868.00       845.00       868.43       871.35                                                                                                                                                                                                                                                                                                                                                                                                                                                                                                                                                                                                                                                                                                                                      |           | $(10^8  \text{m}^3)$ | $(10^8  \text{m}^3)$ | (10° m°)   | (m)      | (m)     | (m)     |             |
| SFY       2.01       1.01       0.67       837.00       822.00       837.97       842.37         WJD       23.00       7.80       13.60       760.00       720.00       760.30       762.80         GPT       64.54       26.62       29.02       630.00       590.00       632.89       638.36         SL       15.93       8.88       3.17       440.00       431.00       445.15       449.45         ST       9.21       4.83       2.87       365.00       353.50       366.73       369.65         DHS       2.77       1.15       1.355       868.00       845.00       868.43       871.35                                                                                                                                                                                                                                                                                                                                                                                                                                                                                                                                                                                                                                                                                                                                                                                                                                       | HJD       | 49.47                | 11.36                | 33.61      | 1140.00  | 1076.00 | 1141.34 | 1145.40     |
| WJD       23.00       7.80       13.60       760.00       720.00       760.30       762.80         GPT       64.54       26.62       29.02       630.00       590.00       632.89       638.36         SL       15.93       8.88       3.17       440.00       431.00       445.15       449.45         ST       9.21       4.83       2.87       365.00       353.50       366.73       369.65         DHS       2.77       1.15       1.355       868.00       845.00       868.43       871.35                                                                                                                                                                                                                                                                                                                                                                                                                                                                                                                                                                                                                                                                                                                                                                                                                                                                                                                                        | DF        | 10.25                | 3.74                 | 4.91       | 970.00   | 936.00  | 975.69  | 977.53      |
| GPT       64.54       26.62       29.02       630.00       590.00       632.89       638.36         SL       15.93       8.88       3.17       440.00       431.00       445.15       449.45         ST       9.21       4.83       2.87       365.00       353.50       366.73       369.65         DHS       2.77       1.15       1.355       868.00       845.00       868.43       871.35                                                                                                                                                                                                                                                                                                                                                                                                                                                                                                                                                                                                                                                                                                                                                                                                                                                                                                                                                                                                                                           | SFY       | 2.01                 | 1.01                 | 0.67       | 837.00   | 822.00  | 837.97  | 842.37      |
| SL     15.93     8.88     3.17     440.00     431.00     445.15     449.45       ST     9.21     4.83     2.87     365.00     353.50     366.73     369.65       DHS     2.77     1.15     1.355     868.00     845.00     868.43     871.35                                                                                                                                                                                                                                                                                                                                                                                                                                                                                                                                                                                                                                                                                                                                                                                                                                                                                                                                                                                                                                                                                                                                                                                             | WJD       | 23.00                | 7.80                 | 13.60      | 760.00   | 720.00  | 760.30  | 762.80      |
| ST     9.21     4.83     2.87     365.00     353.50     366.73     369.65       DHS     2.77     1.15     1.355     868.00     845.00     868.43     871.35                                                                                                                                                                                                                                                                                                                                                                                                                                                                                                                                                                                                                                                                                                                                                                                                                                                                                                                                                                                                                                                                                                                                                                                                                                                                              | GPT       | 64.54                | 26.62                | 29.02      | 630.00   | 590.00  | 632.89  | 638.36      |
| DHS 2.77 1.15 1.355 868.00 845.00 868.43 871.35                                                                                                                                                                                                                                                                                                                                                                                                                                                                                                                                                                                                                                                                                                                                                                                                                                                                                                                                                                                                                                                                                                                                                                                                                                                                                                                                                                                          | SL        | 15.93                | 8.88                 | 3.17       | 440.00   | 431.00  | 445.15  | 449.45      |
|                                                                                                                                                                                                                                                                                                                                                                                                                                                                                                                                                                                                                                                                                                                                                                                                                                                                                                                                                                                                                                                                                                                                                                                                                                                                                                                                                                                                                                          | ST        | 9.21                 | 4.83                 | 2.87       | 365.00   | 353.50  | 366.73  | 369.65      |
| GLQ 0.77 0.51 0.19 719.00 709.00 719.40 722.58                                                                                                                                                                                                                                                                                                                                                                                                                                                                                                                                                                                                                                                                                                                                                                                                                                                                                                                                                                                                                                                                                                                                                                                                                                                                                                                                                                                           | DHS       | 2.77                 | 1.15                 | 1.355      | 868.00   | 845.00  | 868.43  | 871.35      |
|                                                                                                                                                                                                                                                                                                                                                                                                                                                                                                                                                                                                                                                                                                                                                                                                                                                                                                                                                                                                                                                                                                                                                                                                                                                                                                                                                                                                                                          | GLQ       | 0.77                 | 0.51                 | 0.19       | 719.00   | 709.00  | 719.40  | 722.58      |

## 3 Methods

First, this study proposes a storage capacity LR indicator based on sediment accumulation to estimate the WLSV relationship of reservoirs. Second, to identify the impact of the reservoir WSLV relationship on flood control operation, flood regulation calculations are performed at reservoirs with relatively higher LR values using six design flood hydrographs with different return intervals as the typical inflow discharge. Finally, the flood regulation risks are assessed to quantify the response of the scheduling process to the reconstructed WLSV relationship. Figure 2 shows the detailed methods and steps of this study.

Figure 2. Schematic of the framework in this study.

# 3.1 Deriving the reservoir capacity loss volume and rate based on sediment accumulation

In this study, we calculated the reservoir storage capacity LR based on sediment accumulation for each reservoir in the cascade system using the inflow series, sand content in inflow water, and total reservoir capacity during the period of the documented data. Then, we determined the corrected LR for the storage capacity of each reservoir since its construction. Since multi-year average sediment concentration data of inflow water were used instead of annual sediment concentration data, we assumed that the sediment accumulation was spatio-temporally uniform, i.e., the sediment was uniformly distributed at the bottom, and the annual deposition velocity was constant. Then, the LR represents the proportion of multi-year average reservoir storage capacity loss. LR is calculated as follows:

$$LR_{i} = \frac{n_{i} \times w_{i} \times Te_{i}}{\rho \times RC_{i}}$$
(1)

where  $n_i$  is the temporal interval of recorded data for the *i*th reservoir, measured in years;  $RC_i$  is the reservoir capacity, measured in  $10^8$  m<sup>3</sup>;  $w_i$  is the long-term average sediment in the inflow water of the *i*th reservoir, measured in  $10^8$  kg/a;  $\rho$  is the density of deposited sediment, measured in kg/m<sup>3</sup>.  $Te_i$  is the sediment trapping efficiency, and there are multiple detailed methods to estimate its value. Because sediment accumulation processes are complex, we introduced five common empirical models to comparatively evaluate their performance (see Table 2). Hereafter, we used the Brune model to calculate  $Te_i$  according to its applicability in the Wujiang River Basin.

The  $n_i$  is bounded by the service life of reservoirs ( $\leq 100$  years), while  $w_i$  and  $Q_i$  are dynamic variables, evolving with watershed management, reservoir operation, and climate change, thereby limiting the long–term growth of LR.

**Table 2.** Five empirical models used in this study for estimating  $Te_i$ 

| Models                             | Equations                                                                                                                                             |
|------------------------------------|-------------------------------------------------------------------------------------------------------------------------------------------------------|
| Brune                              | $Te_i = 1 - \frac{0.05\alpha}{\sqrt{\Lambda \tau}}$                                                                                                   |
| Brown                              | $Te_i = 1 - \frac{1}{(1 + 0.0021D\frac{RC_i}{A})}$                                                                                                    |
| Gill                               | $Te_{i} = \frac{\left(RC_{i} / Q_{i}\right)^{2}}{0.994701\left(RC_{i} / Q_{i}\right)^{2} + 0.006297\left(RC_{i} / Q_{i}\right) + 0.3 \times 10^{-5}}$ |
| Jothiprakash<br>(coarse sediments) | $Te_{i} = \frac{8000 - 36(RC_{i}/Q_{i})^{-0.78}}{78.85 + (RC_{i}/Q_{i})^{-0.78}}$                                                                     |
| Jothiprakash<br>(medium sediments) | $Te_{i} = \frac{\left(RC_{i} / Q_{i}\right)}{0.00013 + 0.01\left(RC_{i} / Q_{i}\right) + 1.66 \times 10^{-5} \sqrt{RC_{i} / Q_{i}}}$                  |

<sup>\*</sup>  $\alpha$  is the correction factor;  $\Delta \tau$  is the reservoir stagnation time,  $\Delta \tau = RC_i/Q_i$ ;  $RC_i$  represents the reservoir capacity, measured in m³; A, is watershed area, in km²;  $Q_i$  is the multi-year average inflow water volume at the dam site of the ith reservoir,  $10^8$  m³; and D is factor determined by detention time and sediment particle size which varies between 1, 0.1 and 0.046 for coarse, medium and fine sediment respectively.

#### 3.2 Nonlinear fitting of the reservoir WLSV relationship

The reservoir capacity can be represented by a series of discrete water level and storage

data. However, this representation can cause inefficiencies in the algorithmic resolution of the scheduling model. By contrast, quantitative description using mathematical functions can enhance the efficiency of various algorithms used to solve the scheduling model. Therefore, discrete data points of water levels and storages must be fit into a functional relationship. The current literature on the fitting of reservoir WLSV curves (Cao and Liu, 2018; Wang, 2018) combines elementary functions through rational operations to formulate three types of mathematical functions (Eqs. (2)–(4)). In Eqs. (2)–(4), treating the water level (Z) as the explanatory variable caused a relatively poor fit of the WLSV curve. For example, R<sup>2</sup> (the coefficient of determination) was 0.91 at HJD Reservoir in Eq. (3). However, designating the water level as the response variable improved R<sup>2</sup> to 0.99. Thus, we selected the water level (Z) as the response variable.

$$Z = f_1(V) = \alpha_1 V^3 + \alpha_2 V^2 + \alpha_3 V + \alpha_4$$
 (2)

$$Z = f_2(V) = \beta_1 V^{\beta_2} + \beta_3 \tag{3}$$

$$Z = f_3(V) = \gamma_1 e^{\gamma_2 V} + \gamma_3 e^{\gamma_4 V}$$
 (4)

where V is the reservoir storage capacity,  $10^8$  m<sup>3</sup>; Z is the concurrent water level, m;  $f_1$ ,  $f_2$ , and  $f_3$  are the polynomial, power, and exponential functions, respectively.

To assess the goodness-of-fit of nonlinear models  $f_1$ ,  $f_2$ , and  $f_3$  compared with discrete data points of reservoir water levels and storages, we used three typical statistical indices: the coefficient of determination ( $\mathbb{R}^2$ ), sum of squared error (SSE), and root mean square error (RMSE) (Eqs. (5)–(7)).

256 
$$RMSE = \sqrt{\frac{1}{N} \sum_{m=1}^{N} (Z_m - \hat{Z}_m)^2}$$
 (5)

257 
$$R^{2} = 1 - \frac{\sum_{m=1}^{N} (Z_{m} - \hat{Z}_{m})^{2}}{\sum_{m=1}^{N} (Z_{m} - \bar{Z}_{m})^{2}}$$
 (6)

$$SSE = \sum_{m=1}^{N} (Z_m - \hat{Z}_m)^2$$
 (7)

where  $Z_m$  is the observed water level that corresponds to the assessed reservoir capacity  $V_m$ , m;  $\hat{Z}_m$  is the estimated reservoir level, derived from a back extrapolation of the reservoir WLSV curve, and referred to as  $\hat{Z}_m = f(V_m)$ , m;  $\overline{Z}_m$  is the average water level, m; N is the number of sampled discrete data points from the reservoir

WLSV curve to evaluate the performance of the fitted relationship.

## 3.3 Reconstructing the reservoir WLSV curve based on the loss rate

First, the cumulative storage capacity loss over the period with documented data can be estimated based on the corrected storage capacity LR for each reservoir, which is determined in Section 3.1. Second, the total capacity loss is deducted from the original capacity of discrete data points to derive the calibrated water level—storage discrete data points. Third, the optimal lines of  $f_1$ ,  $f_2$ , and  $f_3$  are fitted to model the calibrated discrete data points of water levels and storages for the nine reservoirs using the goodness—of—fit evaluation metrics in Section 3.2 and the reconstructed WLSV curve Z = f'(V).

In this study, we used two approaches to indirectly analyze the suitability of the reconstructed WLSV curve: the water balance equation and DEM. For the water balance equation, the detected change in reservoir water storage from water balance budgets was compared with that from the storage capacity loss. The change in water storage volume of the reservoir over any time period is determined by the inflow water volume, outflow water volume, rainfall, evaporation, and seepage using the water balance equation, i.e., Eq. (8). The summer flood season in the Wujiang River Basin has rainstorms, so the reservoir water storage significantly fluctuates. During this period, inflow and outflow dominate the changes in reservoir water storage, whereas rainfall, evaporation, and seepage have comparatively minimal effects on the closure of the water balance. We explored the measured inflow and outflow from April to August 2023 to estimate the change in reservoir water storage  $\Lambda V$  according to the water balance relation.

$$\Delta V' = V_{\rm in} - V_{\rm out} - V_{\rm ET} - V_{\rm seep} + V_{\rm rain}$$
 (8)

where  $\Lambda V'$  is the change in reservoir water storage, m<sup>3</sup>;  $V_{\rm in}$  is the volume of inflow water into the reservoir, m<sup>3</sup>;  $V_{\rm out}$  is the volume of outflow water from the reservoir, m<sup>3</sup>;  $V_{\rm ET}$  is the volume of evaporated water from the reservoir, m<sup>3</sup>;  $V_{\rm seep}$  is the volume of reservoir leakage water, m<sup>3</sup>;  $V_{\rm rain}$  is the volume of rainfall entering the reservoir surface, m<sup>3</sup>.

For the DEM, we derived discrete data of water levels and storages based on the traditional method using DEM data to quantify the similarity with the reconstructed WLSV curve. In this process, we selected the Copernicus DEM digital elevation model with a resolution of 30 m to create a digital triangular grid, used the ArcGIS platform to extract the surface area for each elevation at 1-m intervals, obtained a series of scatter points that represented the relationship between water levels and surface areas, used the prismatic table method (Eqs. (9)–(10)) to calculate the reservoir storage capacity changes between adjacent water levels, and finally integrated the volume computed by ArcGIS at the initial elevation surface to derive the WLSV curve. This WLSV curve was compared with the reconstructed and designed reservoir WLSV curves.

$$V_{n} = \frac{1}{3} \Delta h \left( S_{n-1} + \sqrt{S_{n-1} S_{n}} + S_{n} \right)$$
 (9)

$$V = \sum_{n=1}^{L} V_n + V_0 \tag{10}$$

where  $V_n$  is the difference in reservoir capacity between two neighboring water levels,  $10^8 \text{ m}^3$ ;  $\Delta h$  is the water level difference between two neighboring water levels, m;  $S_n$  and  $S_{n-1}$  are the water surface areas of the two neighboring water levels,  $m^2$ ; n is the serial number; L is the cumulative number;  $V_0$  is the initial reservoir capacity,  $10^8 \text{ m}^3$ .

## 3.4 Quantifying the response of flood regulation risk to the WLSV curve

To demonstrate that the current reservoir WLSV curve must be rechecked, we selected reservoirs with higher LR values to calculate the flood regulation and assess the resulting flood risks with the reservoir WLSV curve before and after reconstruction. Six design flood hydrographs with return intervals of 200–10,000 years were individually taken as the reservoir inflow discharge to quantify the flood operation risks. The flood operation risk was quantified based on the maximum regulation water level  $(Z^*)$  and the ratio of the number of time periods in which the characteristic water level was surpassed to the total number of time periods ( $\gamma$ ). Flood regulation calculations were conducted using the water balance equation (Eq. (11)). If the generated  $Z^*$  and  $\gamma$  from the reconstructed WLSV curve exceed those from the design WLSV curve currently in use, the continued use of the existing design WLSV curve can

underestimate the risk in flood control and dispatching.

320 
$$V_2 - V_1 = \frac{(Q_1 + Q_2)}{2} \times \Delta t - \frac{(q_1 + q_2)}{2} \times \Delta t$$
 (11)

where  $V_1$  and  $V_2$  are the reservoir storage capacities at the beginning and end of the time period,  $m^3$ , respectively;  $Q_1$  and  $Q_2$  are the inflow discharges at the beginning and end of the time period,  $m^3$ /s, respectively;  $q_1$  and  $q_2$  are the outflow discharges at the beginning and end of the time period,  $m^3$ /s, respectively;  $\Delta t$  is the length of the time period, h.

# 4 Results

## 4.1 Analysis of the total reservoir storage capacity loss rate and volume

Figure 3 illustrates the temporal dynamics of the annual average inflow for each reservoir. In Figure 3, the average annual inflow of reservoirs along the main stream progressively increases from upstream to downstream of the Wujiang River. The HJD Reservoir acts as the primary reservoir in the cascade system and has a low annual average inflow of approximately 100–200 m³/s. DF, SFY, and WJD reservoirs exhibit median annual average inflows of 200–500 m³/s. By contrast, GPT, SL, and ST reservoirs receive high annual average inflow of 500–1000 m³/s. Since the DHS and GLQ reservoirs are in the tributaries of the Wujiang River, they have significantly lower annual average inflows than the main stream reservoirs: their peak annual average inflow was 113.53 m³/s in 2020.

**Figure 3.** Dynamics of annual average inflow of the cascade reservoirs (The periods covered by inflow data varies due to the reservoirs constructed in tributary and downstream is later than those upstream.)

The *Te* of the nine reservoirs in the Wujiang River was calculated using five empirical models; Table 3 shows the results. All models demonstrated consistent trends: upstream reservoirs had higher *Te* than downstream reservoirs, and large reservoirs had higher *Te* than medium-sized reservoirs. Notably, the Gill and Jothiprakash (coarse sediments) models yielded anomalous results at the HJD Reservoir with *Te* values greater than 1 (both were 1.001). The Brown and Jothiprakash (medium sediments) models generated relatively higher *Te* estimates at the HJD, WJD, and GPT reservoirs (large–sized), which approached 0.990, 0.946, and 0.969, respectively. The Brune model produced lower *Te* values (0.957, 0.885, and 0.916) at HJD, WJD, and GPT (large reservoirs) than the other models but higher estimates than the Jothiprakash (medium sediment) model at SFY, ST, and GLQ (medium–sized reservoirs). Thus, we selected the Brune model for calculating *Te*, which demonstrated relatively stable performance in the Wujiang River basin. This aligns with its superior performance over other models in China's Yangtze and Lancang basins in previous investigations (Fu and He, 2007; Tan et al., 2019).

**Table 3.** The *Te* of nine reservoirs in the Wujiang River Basin, as calculated by five tested empirical models.

| Reservoir | Brune | Brown | Gill  | Jothiprakash       | Jothiprakash       |  |
|-----------|-------|-------|-------|--------------------|--------------------|--|
| Reservoir |       |       |       | (coarse sediments) | (medium sediments) |  |
| HJD       | 0.957 | 0.990 | 1.001 | 1.001              | 0.989              |  |
| DF        | 0.856 | 0.922 | 0.955 | 0.930              | 0.899              |  |
| SFY       | 0.648 | 0.659 | 0.761 | 0.725              | 0.604              |  |
| WJD       | 0.885 | 0.946 | 0.973 | 0.953              | 0.932              |  |
| GPT       | 0.916 | 0.969 | 0.989 | 0.977              | 0.962              |  |
| SL        | 0.818 | 0.873 | 0.927 | 0.895              | 0.849              |  |
| ST        | 0.733 | 0.780 | 0.850 | 0.812              | 0.726              |  |
| DHS       | 0.852 | 0.931 | 0.952 | 0.926              | 0.894              |  |
| GLQ       | 0.701 | 0.774 | 0.818 | 0.780              | 0.679              |  |

Table 4 shows the intermediate results of deriving the LR of the water storage capacity for each reservoir in the cascade system. Then, we derived the total reservoir capacity loss attributed to sediment deposition in each reservoir from the reservoir capacity LR, as illustrated in Figure 4. In Figure 4, the SFY Reservoir had the highest LR value (25.02%), followed by the DF and WJD reservoirs with LRs of 12.96% and 7.51%, respectively. The remaining six reservoirs had lower capacity loss rates of 0.32–4.49%. The WJD Reservoir had the highest volume of capacity loss at 172.8 million m³, followed by the DF, HJD, and SFY Reservoirs with losses of 132.9, 69.5, and 50.3 million m³, respectively. The other five reservoirs had lower capacity losses of 3.5–26.9 million m³. Thus, reservoirs with large capacity losses (above 5%) are in the middle and upper reaches of the Wujiang River, which is consistent with the principle that upstream reservoirs prioritize the capture of sand in inflow water.

**Table 4.** Calculation table for the rate of loss of storage capacity

| Reservoir | Time interval of | Sand content of                    | $RC_i$               | $Te_i$ | $w_i$               |
|-----------|------------------|------------------------------------|----------------------|--------|---------------------|
|           | data (year)      | inflow water (kg/ m <sup>3</sup> ) | $(10^8 \text{ m}^3)$ | (%)    | $(10^8 \text{ kg})$ |
| HJD       | 19               | 1.460                              | 49.470               | 0.957  | 53.509              |
| DF        | 23               | 1.130                              | 10.250               | 0.856  | 94.434              |
| SFY       | 18               | 0.609                              | 2.012                | 0.648  | 60.428              |
| WJD       | 23               | 0.982                              | 23.000               | 0.885  | 118.881             |
| GPT       | 14               | 0.127                              | 64.540               | 0.916  | 22.471              |
| SL        | 14               | 0.154                              | 15.930               | 0.818  | 31.370              |
| ST        | 10               | 0.204                              | 9.210                | 0.733  | 51.374              |

| DHS | 15 | 0.228 | 2.765 | 0.852 | 5.390 |
|-----|----|-------|-------|-------|-------|
| GLQ | 13 | 0.229 | 0.774 | 0.701 | 5.341 |

2.0 Loss of capacity -□-- LR Loss of capacity (108 m³) 1.5 HJD DF 1.0 0.5 0.0 SA. CAN 

**Figure 4.** Rate and volume of the storage capacity loss in various reservoirs due to sedimentation accumulation

#### 4.2 Reconstructing the reservoir WLSV curve

Three mathematical function types  $(f_1, f_2, \text{ and } f_3)$  were used to fit the discrete raw water level–storage data points for each reservoir in the Wujiang cascade system. The statistical indices were quantified to evaluate the goodness–of–fit of the regression coefficients. Figure 5 illustrates that function type  $f_2$  with the formula of  $Z = f_2(V) = \beta_1 V^{\beta_2} + \beta_3$  best modeled the relationship between reservoir water level and storage across the nine reservoirs. The goodness–of–fit indices for type  $f_2$  are consistently similar among different reservoirs, which reflects the superior fitting performance of  $f_2$  and its applicability and reliability in depicting WLSV curves. Function types  $f_1$  and  $f_3$  had satisfactory fitting performance in the DF and WJD reservoirs; however, they performed poorly in other reservoirs, which reveals significant discrepancies among different reservoirs. Thus, function types  $f_1$  and  $f_3$  do not fulfill the reliability requirements for all reservoirs and are not sufficiently applicable. Hence, we selected  $Z = f_2(V) = \beta_1 V^{\beta_2} + \beta_3$  as the appropriate fitting function curve for the cascade reservoir system. The original water level–storage volume discrete point data for each reservoir in this cascade system were fitted to

determine the regression coefficients. Then, these parameters were combined with the discrete point data generated by subtracting the storage capacity losses to obtain new parameters for the reconstructed capacity curves of each reservoir. Table 5 shows the regression coefficients in  $f_2$ , which were optimized using the least-squares method.

Figure 5 Goodness-of-fit metrics for the three function types of  $f_1$ ,  $f_2$  and  $f_3$  to be determined

**Table 5.** Power function parameters in the WLSV curve before and after the reconstruction

| Reservoir | $oldsymbol{eta}_1$ | $eta_2$ | $eta_3$ | $oldsymbol{eta_{1}}^{'}$ | $oldsymbol{eta}_{2}^{'}$ | $oldsymbol{eta_3}^{'}$ |
|-----------|--------------------|---------|---------|--------------------------|--------------------------|------------------------|
| HJD       | 42.057             | 0.3597  | 975.20  | 30.723                   | 0.4107                   | 994.92                 |
| DF        | 71.840             | 0.3216  | 826.14  | 30.266                   | 0.4925                   | 889.29                 |
| SFY       | 69.479             | 0.3851  | 751.93  | 34.466                   | 0.5727                   | 798.90                 |
| WJD       | 67.158             | 0.2842  | 599.49  | 35.327                   | 0.3820                   | 649.81                 |
| GPT       | 76.550             | 0.2643  | 408.33  | 58.417                   | 0.3041                   | 432.20                 |
| SL        | 26.977             | 0.4184  | 363.50  | 19.199                   | 0.4939                   | 374.77                 |
| ST        | 39.405             | 0.3491  | 284.12  | 28.707                   | 0.4123                   | 298.69                 |
| DHS       | 78.548             | 0.3285  | 761.95  | 69.373                   | 0.3553                   | 772.59                 |
| GLQ       | 109.790            | 0.3594  | 622.56  | 89.428                   | 0.4318                   | 643.96                 |

<sup>\*</sup>The parameter  $\beta_i$  (i = 1, 2, 3) represents the regression coefficients of the WLSV functional relationship prior to the reconstruction, while the parameter  $\beta_i'$  (i = 1, 2, 3) demotes the coefficients of the reconstructed WLSV function.

Figure 6 shows the reconstructed storage capacity curves against the currently used design capacity curves for the three reservoirs with the most significant capacity LRs. The SFY and DF reservoirs had significant sediment accumulation: the dead storage capacities considerably decreased from 101.2 and 374.0 million m<sup>3</sup> to 50.9 and 241.1 million m<sup>3</sup>, respectively. This reduction must be considered in future reservoir

scheduling, especially for floods near the dead level, due to the capacity loss. By contrast, the WJD Reservoir maintained a relatively large capacity: its dead storage capacity decreased from 780.0 to 599.7 million m<sup>3</sup>. The storage error from the biased WLSV curve can be disregarded at high water levels but requires careful attention at low water levels, particularly near the dead level, since this error significantly affects normal operation.

Figure 6. Reservoir capacity curves before and after reconstruction of nine reservoirs in this study Except for the SFY, DF, and WJD reservoirs, the storage capacity LR of all other reservoirs was below 6%. The capacity curves before and after the reconstruction had median differences. In Figure 6, the dead capacity of the GLQ and ST reservoirs decreased from 50.7 and 483.0 million m³ to 47.2 and 456.1 million m³, respectively. The error in the reservoir storage capacity curve was negligible during normal operations but can become significant when the water level approaches the dead water level. By contrast, the SL, DHS, HJD, and GPT reservoirs exhibited minimal sedimentation and lower errors in their capacity curves, which ensures that normal dispatch operations remain unaffected.

## 4.3 Suitability analysis of the reconstructed reservoir WLSV curve

Figure 7 shows the scatter plot of  $\Lambda V'$ , which is the volume change in reservoir water storage from the water balance method, against  $\Lambda V$ , which is the volume change from the reconstructed capacity curve based on water level fluctuations. For most reservoirs, the scatters of  $\Lambda V'$  and  $\Lambda V$  were close to the diagonal 1:1 lines and showed  $R^2$  values of 0.75–0.98, which indicates the general suitability of the reconstructed capacity curve. The scatters of storage changes in the SFY and ST reservoirs exhibited a disorganized distribution of data points with  $R^2$  of 0.67–0.69. This low fitting performance may be attributed to the inflow data of the SFY Reservoir, which is primarily due to the abrupt changes in measured water level at the dam, irregular fluctuations, and negative values of the inflow.

**Figure 7.** Correlation between the storage changes generated from sediment accumulation and water balance budgets (10<sup>8</sup> m<sup>3</sup>) for nine reservoirs in this study

Because the SFY and DF reservoirs show the most considerable LR values, we used their DEM data to compute various scatter data points for the water level capacity. Then, we compared them with the reconstructed reservoir capacity curves in Section 4.2 (Figure 8). Figure 8 illustrates that the reservoir capacity data points derived from DEM data aligned more closely with the reconstructed reservoir capacity curves than with the existing designs. However, the trends of these curves slightly deviate from the reconstructed curves. Thus, our proposed method for assessing reservoir capacity curves more accurately captures the temporal loss of total reservoir capacity. The calculation process is independent of measured data for the reservoir area topography, so it is both simple and efficient despite some limitations in spatial representativeness. Nonetheless, it lacks spatial representativeness and cannot quantify the sediment siltation across the river sections.

**Figure 8.** Comparison of reservoir capacity curves reconstructed with DEM and sediment accumulation for SFY and DF reservoirs

## 4.4 Analysis of the response of flood regulation risk to the WLSV curve

The flood regulation calculation focused on the SFY and DF reservoirs, which have the highest and second highest capacity LRs, respectively. The current calibrated flood levels for the SFY and DF reservoirs are 842.37 and 977.53 m, respectively. The design flood hydrograph was selected for six return intervals of 200–10,000 years with design frequencies of 0.01%, 0.02%, 0.05%, 0.1%, 0.2%, and 0.5% (Figure 9). The selected design flood hydrograph was conducted with time intervals of 1 h and durations of 170 h. Figure 10 presents the operation risk for flood control characterized by  $Z^*$  and  $\gamma$  that were individually generated from the design and reconstructed reservoir WLSV curves using radar maps. The derived  $Z^*$  values from the flood regulation calculation of the

reconstructed reservoir WLSV curve for both SFY and DF reservoirs significantly increased with decreasing design flood frequencies. For the design flood hydrograph with six return intervals, the reconstructed WLSV curve generated a higher Z\* than the design WLSV curve. The discrepancy between these two Z\* values increased from 0.20 to 6.75 m and from 0.86 to 1.84 m for the SFY and DF reservoirs, respectively, when the return interval increased from 200 to 1000 years. In addition,  $\gamma$  is greater than that quantified by the design WLSV curve. For the design flood hydrograph with a return interval of 10,000 years,  $\gamma$  of the SFY Reservoir increased from 22% with the design WLSV curve to 23% with the reconstructed WLSV curve. Similarly, the proportion of time periods exceeding the normal storage level of the DF Reservoir increased from 18% with the design WLSV curve to 21% with the reconstructed WLSV curve.

The aforementioned results indicate that, according to the prevailing flood control and management principles, both the maximum reservoir water level and the ratio of flood regulation periods surpassing the characteristic water level derived from the existing design capacity WLSV curve are lower than those generated from the reconstructed capacity WLSV curve. Furthermore, underestimation escalates with the increasing magnitude of the design flood. If the storage capacity of reservoirs has been significantly reduced, continuing to use the original design capacity WLSV curve will underestimate the operational risk for flood control. This issue implies a potential risk of water levels surpassing the designated characteristic level during actual operational management of the reservoirs, which puts the reservoir, dam infrastructure, and downstream flood control at risk. Thus, SFY and DF reservoirs must implement the proposed method in this study to timely reconstruct and apply reconstructed capacity curves in future dispatching operations.

Figure 9. Design flood hydrographs for each frequency at the SFY and DF reservoirs

Figure 10. Summary in risk indictors of the flood regulation calculations for the SFY and DF reservoirs

## 5 Discussion

#### 5.1 Reasonableness and uncertainty in the reconstructed reservoir WLSV curve

Although traditional field measurements with surveying and mapping technology yield

highly accurate WLSV curves, this method often has limited implementation due to associated challenges in complex topographic conditions, long duration, and high cost. Consequently, the in-situ WLSV curve is lacking for most reservoirs around the world, with our studied reservoirs included. Thus, we cannot directly quantify the accuracy of the reconstructed WLSV curve in this study. To analyze the reasonableness of the estimated WLSV curve, we indirectly compared it with those derived from the water balance principle (Ahmad et al., 2022) and high-resolution DEM data (Vanthof and Kelly, 2019). The correlation analysis in Section 4.3 indicates that the water balance approach and DEM approach generated much similar WLSV curves to the reconstructed one for each reservoir. Thus, the reconstructed WLSV is more aligned with the actual conditions of each reservoir than the currently used design. In addition, Gui et al. (2025) reconstructed the WLSV curves of the HJD Reservoir using Synthetic Aperture Radar satellite imagery from Sentinel-1 and estimated a capacity loss of 65 million m<sup>3</sup>. Our estimated sediment-induced capacity loss for the same reservoir (69.5 million m<sup>3</sup>) is strongly consistent with that value. To estimate the reservoir storage capacity, there are tradeoffs between the accuracy, the spatial scale of reservoirs, and the suitable degree of explanatory variables considered. As a previous study on the Three Gorges Reservoir (Jia et al., 2021) indicated, the maximum errors in simulated water level and hydropower unit output were 3.0 m and 50×10<sup>4</sup> kW when used WLSV scatter points but decreased to 2.2 m and 29×10<sup>4</sup> kW with the fitted WLSV curve, respectively. These results highlight that uncertainties in WLSV curves can make the reservoir scheduling calculations deviate and affect the operational reliability. In this study, when we focused on specific reservoirs, the fitted WLSV relationship represented by the selected mathematical models achieved higher performance (R<sup>2</sup> >0.98), since the water levels here were directly correlated with the concurrent storage volume. By contrast, a large-scale

approach that estimates storage capacity for multiple reservoirs on a large scale with 16 influencing factors (reservoir morphological parameters, underlying basin conditions, climate types, etc.) performed significantly worse with the highest correlation coefficient < 0.97 (Yuan et al., 2024). Nevertheless, the XGBoost model driven by 16 explanatory factors is suitable for large-scale applications in that study, e.g., to predict reservoir storage capacities at the national scale.

For different reservoirs, there are inevitably significant differences in river characteristics, inflow and sediment conditions, and reservoir boundary shapes. Therefore, previous studies proposed various *Te* formulas to obtain the most applicable ones for specific reservoirs (Ren et al., 2024). To explore suitable empirical Te models for the Wujiang River Basin, we introduced five classic models for comparison. The results indicated that the Gill and Jothiprakash (coarse sediments) models performed poorly and yielded an unrealistic Te of 1.001 at the HJD Reservoir. The Brown and Jothiprakash (medium sediments) models in the WJD Reservoir overestimated Te (0.946 and 0.932, respectively) compared with the reference value of 0.880 from sediment measurements (Li and Jin, 2014). By contrast, the Brune model estimated Te to be 0.885, which closely matched the values derived from in-situ sediment measurements. This validation confirms the suitability of the Brune model for sediment accumulation in the selected reservoir cascade, which aligns with its superior performance over other models in China's Yangtze and Lancang basins in previous investigations (Fu and He, 2007; Tan et al., 2019). Furthermore, the spatial pattern of storage capacity loss volumes and rates is reasonable among reservoirs with different locations in the cascade. Four reservoirs at the middle and upper reaches of the river had larger capacity loss volumes and rates. This result is highly consistent with the conclusion of more severe sediment accumulation in the upper reaches and the priority of the upper reservoir to trap sand (Brush, 1989; Yuan et al., 2022). This fundamental pattern indirectly offers another evidence of the reasonableness of the reconstructed WLSV curves. In summary, our study contributes a sediment accumulation-based reservoir capacity loss framework to reliably estimate WLSV relationships for multiple reservoirs at a larger scale.

## 5.2 Implications of the study for reservoir operation strategies

This study has revealed the potential impacts of sediment accumulation on the accuracy of reservoir WLSV curves and flood control operations and offers the following implications for reservoir management. Since current design WLSV curves underestimate flood regulation risks, reservoir operations should strengthen sediment accumulation monitoring and conduct periodic recalibrations of WLSV curves for high LR reservoirs as mandated by the Code for Reservoir Hydrologic and Sediment Survey (2006) to mitigate the negative impacts on operational efficiency.

For reservoirs that have severely lost capacity due to sediment accumulation, such as those in the upper–middle Wujiang River, decision–makers can increase sediment discharge by scientifically regulating the reservoir discharge hydrograph to fully utilize the propagation time difference between flood peaks and sediment peaks during flood seasons, while ensuring flood control safety (Ren et al., 2021). Simultaneously, sediment reduction operations should be implemented by optimizing the reservoir water level drawdown process in coordination with incoming water and sediment conditions, thereby promoting scouring of sediment accumulation (Wang et al., 2016). The reservoirs that storage loss is severe and the sediment accumulation is relatively coarse (making scouring difficult), comprehensive measures such as mechanical dredging and engineering remediation should be promptly implemented to partially recover the effective storage capacity lost to sediment accumulation.

## 5.3 Limitations of the proposed framework and outlooks for future studies

Apart from sediment accumulation, other factors such as river diversion and sand mining can change the reservoir storage capacity. In this study, we considered sediment accumulation as the most important factor affecting the storage capacity, while we ignored other factors. This situation is practical for the selected cascade reservoir system in the Wujiang River. If the framework is applied to other reservoirs, the applicability of this assumption should first be carefully diagnosed. Therefore, the framework for reconstructing WLSV relationships is available under relatively strict constraints. Furthermore, the proposed framework has one potential limitation that

should be improved in future studies. Specifically, we assumed that the temporal velocity and spatial distribution of sediment accumulation were uniform using the ratio of the reservoir storage capacity to the incoming flow (*Te*), which was defined in the mathematical relationship of Eq. (2), to simplify the calculation of the storage capacity loss rate. In reality, dams and reservoirs are complex depositional systems for sediment, which vary with the reservoir bottom shapes, river discharges, total volumes of stored water, sediment loads and textures, and universal depositional models (Sedláček et al., 2022). Thus, the next attempt should consider detailed sedimentation processes in reservoirs. We advise the future studies to couple physics-based hydrological models with reservoir operation models (Dong et al., 2022), deep learning algorithms (Cao et al., 2020), and more publicly available explanatory factors, including those extracted from remote sensing missions (Bonnema and Hossain, 2019; Guan et al., 2021), with more flexible non-linear changes in sedimentation, to reduce the complexity of reconstructing the reservoir WLSV relationship, particularly at large scales.

## 6 Conclusions

As intensified climate change and human activities more strongly affect the hydrological process and water balance of reservoirs, the representation of reservoir WLSV curves must be improved to develop more useful flood control adaptation measures. In this study, we contributed a framework to reconstruct reservoir WLSV curves by first constructing the reservoir capacity loss rate (LR) indicator from sediment accumulation. Then, we performed flood regulation calculations of individual reservoirs using six design flood hydrographs with six return intervals of 200-10,000 years as the reservoir inflow to elucidate the response of flood control operation to the reconstructed WLSV curve. Finally, we quantified the flood control risk using the maximum flood control water level ( $Z^*$ ) and the number of periods when the design flood level ( $\gamma$ ) was exceeded. We implemented the established framework on a cascade system of nine reservoirs in the Wujiang River in China. The main conclusions are as follows:

- (1) The results in this study demonstrated that the framework of reservoir storage loss volumes and rates induced by sediment accumulation can reconstruct a suitable reservoir WLSV relationship.
  - (2) The Suofengying, Dongfeng, and Wujiangdu reservoirs, located in the upper reaches of the Wujiang mainstream, had more serious sediment accumulation, which significantly changed the capacity, with LR values over 25.02%, 12.96%, and 10%, respectively. The higher correlation between the water storage changes generated from reconstructed WLSV curves in this study and those based on the water balance and DEM data demonstrates that the estimated WLSV curve with sediment accumulation is suitable for use.
  - (3) The current design WLSV curve for flood regulation calculations underestimated the maximal water level during flood regulation ( $Z^*$ ) and the duration of regulation beyond the characteristic water level ( $\gamma$ ). The underestimation increases with the return period of the design flood. For the Suofengying and Dongfeng reservoirs, the underestimation of  $Z^*$  increased from 0.20 and 0.86 m for the 200–year return period to 6.75 and 1.84 m for the 10,000–year return period. Thus, when the storage capacity significantly decreases, continued use of the existing design WLSV curve may significantly underestimate the flood regulation risks and pose potential safety hazards to the reservoir and downstream flood protection objects.

## **Declaration of competing interest**

- The authors declare that they have no known competing financial interests or personal
- relationships that could have appeared to influence the work reported in this paper.

# 632 Data availability

Data will be made available upon request (qiumeima@ncepu.edu.cn).

#### **Author contributions**

YZ, QM and ZD designed the study. QM, CX, ZD, and CYX developed the models, with QM and CX implementing them. CX drafted the manuscript in close collaboration with YZ, QM and CX contributed to the data curation. Throughout the study period, all the authors engaged in discussions regarding the results, provided critical feedback, and

approved the final version of the paper.

## Acknowledgments

- We are very grateful to the reviewers for their input, which significantly improved this
- paper.

640

# 643 Financial support

- This study was funded by the National Natural Science Foundation of China (52109016,
- U2240201), and the Fundamental Research Funds for the Central Universities
- (2023MS075, 2024JC003). ZD acknowledges the support from the Joint China-
- Sweden Mobility Grant funded by the Swedish Foundation for International
- Cooperation in Research and Higher Education (STINT, CH2019-8250). Additionally,
- we are very grateful to the editors and anonymous reviewers for their valuable
- comments, which cloud greatly improve the quality of the paper.

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
