# Peer review of "Reconstruction of the reservoir water level-storage volume"

_EGUsphere, 2025_

## Referee Comment (RC2)

**Summary:**

This paper proposes a method to reconstruct the water level–storage (WLS) relationship of reservoirs by estimating storage capacity loss due to sediment accumulation. The method is applied to a cascade of nine reservoirs along China's Wujiang River, and the reconstructed WLS curves are validated using water balance analysis and DEM-based surface data. Overall, the framework presented in this study provides useful insights and offers practical value for reservoir management to a certain extent. However, the manuscript would benefit from further refinement to improve its clarity, logical flow, and rigor before it can be considered for publication. In particular, some sections of the paper would benefit from a smoother transition between sentences and a clearer presentation of the methodology and results. Below, I outline specific comments and suggestions for the authors' consideration.

**Major Comments:**

1. I recommend that the authors consider adjusting or restructuring the paragraphs in the Introduction section. The logical flow and overall storyline of the Introduction are currently not entirely smooth. For example, the authors state that "Estimating the storage capacity loss induced by sediment accumulation and reconstructing the WLS curve provides a new direction for the second category of methods," but no clear background or justification is provided to connect this statement with the earlier discussion. Prior sentences only mention that sediment accumulation leads to reservoir storage loss, without adequately building the context for why reconstructing the WLS curve becomes a new methodological direction. Additional examples of logical inconsistencies are noted in my minor comments section.

2. I suggest that the authors add appropriate references throughout the manuscript, particularly when providing background information in the Study Area and Data section (e.g., indicating the sources of the data) and in the Methodology section. Providing references would enhance the transparency and credibility of the study. In addition, it would be helpful if the authors could clarify the rationale behind the selection of the study area and the specific reservoirs.

3. For Equations (1) and (2), the authors make certain assumptions to simplify the derivation. Given that the loss rate (LR) is a key indicator in this manuscript, I suggest that the authors provide more detailed explanations of these assumptions for the benefit of the readers. In particular, it would be helpful to clarify why these assumptions were made, how they affect the methodology, and whether there are existing references that support or justify them.

4. I suggest that the authors consider creating a table listing all abbreviations used throughout the manuscript for easier reference. Additionally, while the abbreviations for each reservoir are defined and used in the figures and text, the writing in Section 4.1 inconsistently switches between full names and abbreviations. For consistency and to improve readability, I recommend using the reservoir abbreviations throughout the text, especially since the figures present the abbreviations as well.

5. The discussion section should be strengthened and expanded. Specifically, the authors could provide a more detailed comparison of their WLS curve reconstruction method with other existing approaches. Additionally, the current discussion (Lines 491–499) may not be sufficient to fully demonstrate the reasonableness of the reconstructed WLS curves. I encourage the authors to elaborate further on the topic of "Reasonableness and uncertainty in reconstructed reservoir WLS curves".

6. Could the authors elaborate on the potential policy implications of this study? How can decision-makers apply these findings to real-world water resource management?

**Minor Comments:**

1. Lines 60 – 63; and 69: Please add appropriate references.

2. Lines 102 – 105: Consider rewriting these sentences for clarity, as the meaning feels vague based on the previous paragraph. In the earlier paragraph, the authors mention that the first method involves topographic surveys to estimate reservoir storage capacity. However, these lines suggest that most reservoirs have not yet conducted storage capacity estimations. Please clarify — do you mean that most methods have not directly estimated storage capacity?

3. Lines 114 - 116: Please add appropriate references.

4. Lines 116 - 118: The logical flow is not smooth; no prior background about flood regulation has been introduced. Please consider revising for better connection and clarity.

5. Line 133: What is meant by "flood prevention operation"? Is this intended to mean "flood control operation"? Please clarify.

6. Line 151: A period is missing after "Fig. 1."

8. Line 171 and Figure 1: Should "GLT" be corrected to "GLQ"? Please double-check.

7. Line 223: Again, a supporting reference is needed here.

8. Lines 239 – 240: Please ensure the formatting of symbols is consistent with the rest of the manuscript.

9. Line 270: Please specify a unit for "$V_{rain}$."

10. The symbol "i" is used with different meanings across equations. Please consider using distinct symbols to avoid confusion and ensure consistency.

11. Line 311: "Figure. 3" should be corrected to "Fig. 3." Please ensure consistent formatting of figure references throughout the manuscript.

12. line 411: "Section 3.2" should be "Section 4.2" ?

13. Line 487: Please add a space before "On the other..."

14. Lines 507 – 509: Please double-check these sentences for accuracy and consider rewriting them for clarity.

---

## Author Comment (AC1)

Reviewer #1:

The study proposes a capacity loss rate (LR) index to reconstruct the water level-storage (WLS) relationship of reservoirs, utilizing measured water/sediment data and operational records. The suitability of the proposed method is evaluated through water balance principles and comparison with traditional approaches. Subsequently, the impact of reconstructed WLS curves on reservoir flood control and operation is quantified using a flood regulation algorithm with varying design inflow scenarios. The experimental design from WLS curve reconstruction to flood operation impact assessment is systematic. While the proposed methods offer practical values for reservoir management, issues remain in language expression and method application. Therefore, I recommend Major Revision for this manuscript. The detailed comments are as follows.

**Reply:** Thanks for your time and invaluable comments, which have substantially improved the study. Based on your comments, we have carefully revised the paper. Please find the detailed point-by-point response to the reviewers' comments in the attachment. The comments have been replied one by one in this document.

The line numbers in the responses below refer to the revised manuscript, while the line numbers in the comments refer to the original manuscript. We hope the revised manuscript can meet the publication standard of the Hydrology and Earth System Sciences at this time.

Comment 1. The terms "Fig." and "Figure" are used interchangeably in the manuscript. Please standardize to one format for consistency. Additionally, proofreading by a native English speaker should be conducted to improve overall language quality.

**Reply:** Thanks for your valuable comment regarding the language in the manuscript. We have standardized all references to "Figure" (e.g., Figure 1). Further, we engaged a native English speaker to polish the manuscript, improving both language and organization.

Comment 2. This study uses the sediment deposition rate (Te) to calculate LR. It is recommended to introduce the concept and development history of Te in Introduction Section and reduce the examples of traditional methods to better highlight the study's core methodology.

**Reply:** Thanks for your valuable comment. We have added a brief introduction to the *Te* in the Introduction Section and listed typical approaches for calculation its values, along with corresponding references (e.g., Brown, 1944; Moragoda et al., 2023; Ren et al., 2024, …etc.). Please see Lines 115-123 in the revised manuscript.

Newly cited references (highlighted in red in the manuscript):

Brown, C. B.: Discussion of sedimentation in reservoirs by B. J. Witzig, Proceedings of the American Society of Civil Engineers, 69, 1493–1500, 1944.

Moragoda, N., Cohen, S., Gardner, J., Muñoz, D., Narayanan, A., Moftakhari, H., and Pavelsky, T. M.: Modeling and Analysis of Sediment Trapping Efficiency of Large Dams Using Remote Sensing, Water Resources Research, 59, e2022WR033296, https://doi.org/10.1029/2022WR033296, 2023.

Ren, S., Gao, Y., Wang, W., Zhou, Y., and Zhao, H.: Estimating Sediment Trap Efficiency of Flood Events During Flood Season in the Three Gorges Reservoir, Water Resources Research, 60, e2023WR036975, https://doi.org/10.1029/2023WR036975, 2024.

Tan, G., Chen, P., Deng, J., Xu, Q., Tang, R., Feng, Z., and Yi, R.: Review and improvement of conventional models for reservoir sediment trapping efficiency, Heliyon, 5, e02458, https://doi.org/10.1016/j.heliyon.2019.e02458, 2019.

Comment 3. For 2.1 Section of the study area, in Line 151, a period is missing after "Fig. 1". Please revise throughout the manuscript. In Line 152, the phrase "the Wujiang River Basin has aggravated the interception of sediment" should be reworded to "sediment interception in the basin has increased".

**Reply:** Thanks for your valuable comment. We have added the missing period after "Fig. 1" in Line 162 and the phrase in Line 163-164 has been revised to "sediment interception in the basin has increased" to clarify the intended meaning.

Comment 4. In Methods Section, since the Brune model is an empirical model based on U.S. reservoir siltation survey data, its applicability to reservoir siltation calculation in China may be limited. It is suggested to add a discussion on the model's applicability, supported by relevant literature.

**Reply:** Thanks for your valuable comment. We introduced four other empirical models commonly used to compare them with the selected Brune model, in Section 3.1 (Lines 222-226). The results shows that the Brune model demonstrates better applicability while the Gill and Jothiprakash (coarse sediments) models yield clearly unreasonable results (notably $Te = 1.001 > 1$ at HJD reservoir), the Brown model overestimates $Te$ in three large reservoirs (HJD, WJD, GPT; 0.946-0.990), and the Jothiprakash (medium sediments) model underestimates $Te$ in three small reservoirs (SFY, ST, GLQ; 0.604-0.736). Furthermore, the $Te$ value derived from the Brune model for WJD Reservoir (0.885) closely matches the result determined by Li and Jin (2014) using hydrological station data (0.88). Fu and He (2007) and Tan et al. (2019) additionally confirm the Brune model's suitability for reservoirs in the Yangtze and Lancang River basins. Please see the details in Lines 539- 547 of the revised manuscript.

Newly cited references (highlighted in red in the manuscript):

Fu, K. and He, D.: Analysis and prediction of sediment trapping efficiencies of the reservoirs in the mainstream of the Lancang River, Chin. Sci. Bull., 52, 134–140, https://doi.org/10.1007/s11434-007-7026-0, 2007.

Li, J. and Jin, Z.: Studying the reservoir sedimentation problem in upper Yangtze River by trap efficiency method, 416–420, 2014.

Tan, G., Chen, P., Deng, J., Xu, Q., Tang, R., Feng, Z., and Yi, R.: Review and improvement of conventional models for reservoir sediment trapping efficiency, Heliyon, 5, e02458, https://doi.org/10.1016/j.heliyon.2019.e02458, 2019.

Comment 5. In 3.1, according to Eq. 1, as ni in the denominator increases, the reservoir capacity loss rate LRi also increases. This implies that LR grows over time, suggesting an unrealistic scenario where reservoirs would silt up indefinitely. Please clarify the constraints of this equation to reflect real-world limitations.

**Reply:** Thanks for your valuable comment. For physical constraints: According to Hua (2014), the sediment measurements at WJD Reservoir in 1989 and 2012 showed the sediment accumulation of 210 million m³ and 186.8 million m³, respectively. Our calculated result (180.3 million m³, LR=7.84%) aligns closely with this value. Further analysis reveals that the sediment accumulation of WJD Reservoir has slowly decreased in recent years due to enhanced sediment retention by upstream reservoirs and the "Returning Farmland to Forests and Grasslands" policy. In addition, sediment is susceptible to being washed away due to the high variability of the reservoir water level. These real-world factors collectively constrain the continuous growth of LR.

For mathematical parameter constraints: In Eq. (1), $n_i$ in the denominator represents the temporal interval of recorded data, which is bounded by the service life of reservoirs (typically ≤ 100 years), thereby imposing a theoretical upper limit on LR. Furthermore, the parameters $Q_i$ (the multi-year average inflow water volume at the dam site) and $w_i$ (the long-term average sediment contained in the inflow water) are dynamic variables. Their values evolve with watershed management practices, reservoir operation strategies, and climatic changes, further curbing the long-term growth trend of LR.

In summary, Eq. (1) inherently incorporates constraints from both physical processes and mathematical parameter, ensuring that LR cannot increase indefinitely as $n_i$ increases.

Comment 6. The study establishes Brune model to calculate Te, but multiple approaches exist for this purpose. It is recommended to compare the results with those from other models in the Methods section to strengthen methodological robustness.

**Reply:** Thanks for your valuable comment. Thanks for your valuable comment. We introduced four other empirical models commonly used to compare them with the selected Brune model, in Section 3.1 (Lines 222- 226). The results shows that the Brune model demonstrates better applicability, while the Gill and Jothiprakash (coarse sediments) models yield clearly unreasonable results (notably $Te = 1.001 > 1$ at HJD reservoir), the Brown model overestimates $Te$ in three large reservoirs (HJD, WJD, GPT; 0.946-0.990), and the Jothiprakash (medium sediments) model underestimates $Te$ in three small reservoirs (SFY, ST, GLQ; 0.604-0.736). Please see Lines 345- 356 in the revised manuscript.

**Comment 7.** The reservoir exhibits complicated morphometry, with significant variations in water depth and width. In the current study, several reservoirs (e.g., Hongjiadu and Goupitan) also cover extensive surface areas. On the other hand, the authors employ mathematical functions with exponential, polynomial and power function forms in Eqs. (3-5) to estimate reservoir water storage changes. Please address potential errors induced by the simplified estimation formula. Could the accuracy of the WLS curve be improved by subdividing the reservoir into smaller sections based on morphometry and summing their contributions?

**Reply:** Thanks for your constructive comment. Although summing several subsections could improve accuracy, it would require detailed bathymetric data (e.g., high-resolution DEMs of reservoir areas, measured underwater topography, and cross-sectional hydrological profiles), which is indeed unavailable in the present study. We use various mathematical function types including exponential, polynomial and power functions in the quantitative representation of WLSV curve to reduce modeling errors. Additionally, we plan to explore more refined methods in future studies.

**Comment 8.** In Eqs. (3-5), the water level ($Z$) is treated as the response variable, while the reservoir water storage is the explanatory variable, Yet, the reservoir WLS curve reconstruction is evaluated based on the water storage variable. This inconsistency needs clarification.

**Reply:** Thanks for your valuable comment. In Eqs. 7-9, when water level ($Z$) is treated as the explanatory variable, the function exhibits poor fitting performance for the WLSV curve. For instance, the $R^2$ is 0.91 at HJD Reservoir. However, the $R^2$ significantly improves to 0.99 when water level ($Z$) is instead designated as the response variable. Consequently, we selected water level ($Z$) as the response variable in our study. For practical applications, the mathematical function can be inverted to derive water level ($Z$) as the explanatory variable. Please see Lines 241-245 in the revised manuscript.

**Comment 9.** In Figure 1, the abbreviation for the "GLT" reservoir should be corrected to "GLQ" (Geliqiao).

**Reply:** Thanks for your comment. The abbreviation for "Geliqiao Reservoir" in Figure 1 has been corrected to "GLQ", with consistency verified throughout the manuscript. Please see Line 181 in the revised manuscript.

**Comment 10.** The Discussion Section should include a comparison of your WLS curve construction method with other existing approaches. Additionally, please elaborate on how the accuracy of WLS curve impacts the reliability of the water storage estimation.

**Reply:** Thanks for your valuable comment. We have supplemented the comparison of our WLSV curve construction method with other approaches. Gui (2025) reconstructed the WLSV curves of HJD Reservoir using Sentinel-1 SAR data, estimating capacity losses of 65 million m³. Our estimated sediment-induced capacity loss for the same reservoir (69.5 million m³) shows strong consistency with these results.

Furthermore, Pei (2021) validated the rationality of simplified mathematical functions (third-order polynomial fitting) for WLSV curve reconstruction by combining terrestrial 3D laser scanning and unmanned vessel bathymetry, which aligns with our methodological framework.

Furthermore, uncertainties in WLSV curves potentially affect the estimation accuracy of water level and operational reliability. Previous study on the Three Gorges Reservoir (Jia et al., 2021) shows that the reasonableness and uncertainty of WLSV curves critically affect operational reliability: the maximum errors in simulated water level and hydropower output were 3.0 m and $50\times10^4$ kW when using WLSV scatter points, respectively, but decreased to 2.2 m and $29\times10^4$ kW with fitted WLSV curve. That results demonstrate that uncertainties in WLSV curves can lead to deviations in reservoir scheduling calculations, thereby impacting power generation efficiency and flood risk management. This analysis has been added to Lines 513-519 and Lines 525-532 in the revised manuscript.

Newly cited references (highlighted in red in the manuscript):

Gui, X., Ma, Q., Li, J., Duan, Z., Xiong, L., and Xu, C.-Y.: Reconstructing Reservoir Water Level-Area-Storage Volume Curve Using Multi-source Satellite Imagery and Intelligent Classification Algorithms, Water Resour Manage, https://doi.org/10.1007/s11269-025-04205-7, 2025.

Pei, J., Yao, C., Deng, Z., Ren, Z., and Yang, Y.: The Application of 3D Laser Scanning and Unmanned Ship Sounding in the Reexamination of Reservoir Storage Capacity, IOP Conf. Ser.: Earth Environ. Sci., 719, 042052, https://doi.org/10.1088/1755-1315/719/4/042052, 2021.

Jia, B., Zhou, J., Chen, X., Tian, M., and Zhang, Y.: Fitting reservoir stage-capacity curves and its application in reservoir operation, Journal of Hydroelectric Engineering, 40, 89–99, https://doi.org/DOI: 10.11660/slfdxb.20210209, 2021.

Comment 11. The abbreviations in the text and figures should be independent of each other. It is recommended that the corresponding vocabulary of the reservoir name abbreviations should also be clearly stated in the data methods in the manuscript text.

**Reply:** Thanks for the useful comment. A comprehensive list of abbreviations has been added, explicitly defining the abbreviation conventions for reservoir names. The entire manuscript has been revised to consistently use abbreviations (with full names introduced at first mention) in both text and figure.

**Table 1.** Full names used in articles and their corresponding abbreviations

| Full name | Abbreviation |
| --- | --- |
| water level-storage volume | WLSV |
| loss rate | LR |
| Hongjiadu | HJD |
| Dongfeng | DF |
| Suofengying | SFY |
| Wujaingdu | WJD |
| Goupitan | GPT |
| Silin | SL |
| Shatuo | ST |
| Dahuashui | DHS |
| Geliqiao | GLQ |

---

## Author Comment (AC2)

Reviewer #2:

This paper proposes a method to reconstruct the water level–storage (WLS) relationship of reservoirs by estimating storage capacity loss due to sediment accumulation. The method is applied to a cascade of nine reservoirs along China's Wujiang River, and the reconstructed WLS curves are validated using water balance analysis and DEM-based surface data. Overall, the framework presented in this study provides useful insights and offers practical value for reservoir management to a certain extent. However, the manuscript would benefit from further refinement to improve its clarity, logical flow, and rigor before it can be considered for publication. In particular, some sections of the paper would benefit from a smoother transition between sentences and a clearer presentation of the methodology and results. Below, I outline specific comments and suggestions for the authors' consideration.

**Reply:** Thank you for your efforts and constructive comments, which have significantly improved the quality of the manuscript. We have provided detailed point-by-point responses to the reviewers' comments and meticulously revised the manuscript. The response document is provided as an attachment, where the line numbers in replies refer to the revised manuscript, while those in the comments refer to the original manuscript. We hope the revised manuscript now meets the publication standard of the Hydrology and Earth System Sciences.

Comment 1. I recommend that the authors consider adjusting or restructuring the paragraphs in the introduction section. The logical flow and overall storyline of the Introduction are currently not entirely smooth. For example, the authors state that "Estimating the storage capacity loss induced by sediment accumulation and reconstructing the WLS curve provides a new direction for the second category of methods," but no clear background or justification is provided to connect this statement with the earlier discussion. Prior sentences only mention that sediment accumulation leads to reservoir storage loss, without adequately building the context for why reconstructing the WLS curve becomes a new methodological direction. Additional examples of logical inconsistencies are noted in my minor comments section.

**Reply:** Thanks for your valuable comment. We have restructured the paragraph of the Introduction Section in Lines 106-128. The possible effects of sediment on reservoirs, especially on flood control operation, are introduced first. Then, the development of $Te$ is described to better highlight the core methodology. Finally, the shortcomings investigated by current studies are summarized, such as, few studies have investigated the validity of reconstructing the WLSV curve based on sediment accumulation, and there has been insufficient attention to the practical effectiveness of the reconstructed curve. Therefore, Using the estimation of storage capacity loss induced by sediment accumulation to reconstruct the WLSV curve provides a new direction for the second category of methods for reconstructing WLSV curves.

Newly cited references (highlighted in red in the manuscript):

Brown, C. B.: Discussion of sedimentation in reservoirs by B. J. Witzig, Proceedings of the American Society of Civil Engineers, 69, 1493–1500, 1944.

Moragoda, N., Cohen, S., Gardner, J., Muñoz, D., Narayanan, A., Moftakhari, H., and Pavelsky, T. M.: Modeling and Analysis of Sediment Trapping Efficiency of Large Dams Using Remote Sensing, Water Resources Research, 59, e2022WR033296, https://doi.org/10.1029/2022WR033296, 2023.

Ren, S., Gao, Y., Wang, W., Zhou, Y., and Zhao, H.: Estimating Sediment Trap Efficiency of Flood Events During Flood Season in the Three Gorges Reservoir, Water Resources Research, 60, e2023WR036975, https://doi.org/10.1029/2023WR036975, 2024.

Tan, G., Chen, P., Deng, J., Xu, Q., Tang, R., Feng, Z., and Yi, R.: Review and improvement of conventional models for reservoir sediment trapping efficiency, Heliyon, 5, e02458, https://doi.org/10.1016/j.heliyon.2019.e02458, 2019.

Comment 2. I suggest that the authors add appropriate references throughout the manuscript, particularly when providing background information in the Study Area and Data section (e.g., indicating the sources of the data) and in the Methodology section. Providing references would enhance the transparency and credibility of the study. In addition, it would be helpful if the authors could clarify the rationale behind the selection of the study area and the specific reservoirs.

**Reply:** Thanks for your valuable comment. We have added the corresponding references (Wu et al., 2018; Yuan et al., 2022) to the Study Area Section when introducing the background of the Wujiang River Basin. In the Data Section, the water and sediment data utilized in this study were obtained from a professional engineering organization under a data-sharing agreement. We conducted rigorous quality control for the data, involving manual verification and rectification of anomalous and erroneous data. Please see Lines 187-190 in the revised manuscript.

Newly cited references (highlighted in red in the manuscript):

Wu, X., Xiang, X., Chen, X., Zhang, X., and Hua, W.: Effects of cascade reservoir dams on the streamflow and sediment transport in the Wujiang River basin of the Yangtze River, China, Inland Waters, 8, 216–228, https://doi.org/10.1080/20442041.2018.1457850, 2018.

Yuan J., Chen W., Yang C., and Xiong M.: Study on Sediment Retention and Reduction of Reservoirs in Wujiang River Basin, JWRR, 11, 249–259, https://doi.org/10.12677/JWRR.2022.113027, 2022.

Comment 5. For Equations (1) and (2), the authors make certain assumptions to simplify the derivation given that the loss rate (LR) is a key indicator in this manuscript, I suggest that the authors provide more detailed explanations of these assumptions for the benefit of the readers. In particular, it would be helpful to clarify why these assumptions were made, how they affect the methodology, and whether there are existing references that support or justify them.

**Reply:** Thanks for your valuable comment. This study adopted the multi-year average sediment concentration data of inflow water for each reservoir. The results calculated based on these data reflect the multi-year average reservoir capacity loss, thus we assume that the sediment is uniformly distributed in the bottom and the deposition velocity is the same every year. If detailed annual sediment concentration data of reservoir inflows were available, the methodology proposed in this study could also be applied to estimate annual reservoir capacity loss. The relevant explanations have been added to Lines 211-214 in the revised manuscript.

Comment 4. I suggest that the authors consider creating a table listing all abbreviations used throughout the manuscript for easier reference. Additionally, while the abbreviations for each reservoir are defined and used in the figures and text, the writing in Section 4.1 inconsistently switches between full names and abbreviations. For consistency and to improve readability, I recommend using the reservoir abbreviations throughout the text, especially since the figures present the abbreviations as well.

**Reply:** Thanks for your comment. A comprehensive list of abbreviations has been added, explicitly defining the abbreviation conventions for reservoir names. The entire manuscript has been revised to consistently use abbreviations (with full names introduced at first mention) in both text and figure.

**Table 1.** Full names used in articles and their corresponding abbreviations

| Full name | Abbreviation |
| --- | --- |
| water level-storage volume | WLSV |
| loss rate | LR |
| Hongjiadu | HJD |
| Dongfeng | DF |
| Suofengying | SFY |
| Wujaingdu | WJD |
| Goupitan | GPT |
| Silin | SL |
| Shatuo | ST |
| Dahuashui | DHS |
| Geliqiao | GLQ |

Comment 5. The discussion section should be strengthened and expanded. Specifically, the authors could provide a more detailed comparison of their WLS curve reconstruction method with other existing approaches. Additionally, the current discussion (Lines 491–499) may not be sufficient to fully demonstrate the reasonableness of the reconstructed WLS curves. I encourage the authors to elaborate further on the topic of "Reasonableness and uncertainty in reconstructed reservoir WLS curves".

**Reply:** Thanks for your valuable comment. We have supplemented the comparison of our reconstruction method for WLSV curve with other approaches. Gui (2025) reconstructed the WLSV curves of HJD Reservoir using Sentinel-1 SAR data, estimating capacity losses of 65 million m³. Our estimated sediment-induced capacity loss for the same reservoir (69.5 million m³) shows strong consistency with these results. Furthermore, Pei (2021) validated the rationality of simplified mathematical functions (third-order polynomial fitting) for WLSV curve reconstruction by combining terrestrial 3D laser scanning and unmanned vessel bathymetry, which aligns with our methodological framework.

Furthermore, uncertainties in WLSV curves potentially affect the estimation accuracy of water level and operational reliability. Previous study on the Three Gorges Reservoir (Jia et al., 2021) shows that the reasonableness and uncertainty of WLSV curves critically affect operational reliability: the maximum errors in simulated water level and hydropower output were 3.0 m and $50 \times 10^4$ kW when using WLSV scatter points, respectively, but decreased to 2.2 m and $29 \times 10^4$ kW with fitted WLSV curve. That results demonstrate that uncertainties in WLSV curves can lead to deviations in reservoir scheduling calculations, thereby impacting power generation efficiency and flood risk management. Relevant content has been added to Lines 513-519 and Lines 525-532 in the revised manuscript.

Newly cited references (highlighted in red in the manuscript):

Gui, X., Ma, Q., Li, J., Duan, Z., Xiong, L., and Xu, C.-Y.: Reconstructing Reservoir Water Level-Area-Storage Volume Curve Using Multi-source Satellite Imagery and Intelligent Classification Algorithms, Water Resour Manage, https://doi.org/10.1007/s11269-025-04205-7, 2025.

Pei, J., Yao, C., Deng, Z., Ren, Z., and Yang, Y.: The Application of 3D Laser Scanning and Unmanned Ship Sounding in the Reexamination of Reservoir Storage Capacity, IOP Conf. Ser.: Earth Environ. Sci., 719, 042052, https://doi.org/10.1088/1755-1315/719/4/042052, 2021.

Jia, B., Zhou, J., Chen, X., Tian, M., and Zhang, Y.: Fitting reservoir stage-capacity curves and its application in reservoir operation, Journal of Hydroelectric Engineering, 40, 89–99, https://doi.org/DOI: 10.11660/slfdxb.20210209, 2021.

**Comment 6. Could the authors elaborate on the potential policy implications of this study? How can decision-makers apply these findings to real-world water resource management?**

**Reply:** Thanks for your valuable comment. In the newly added Section 5.2 (Lines 557-569), we analyze the policy implications arising from the study's findings and their insights for reservoir management, as follows: As current design WLSV curves underestimate flood regulation risk, reservoir operations should strengthen sediment accumulation monitoring and conduct periodic recalibration of WLSV curves for high LR reservoirs as mandated by Code for Reservoir Hydrologic and Sediment Survey (2006), to mitigate the negative impacts on the operational efficiency. Furthermore, reservoirs in the upper-middle Wujiang River suffer serious capacity loss from sediment accumulation. For such reservoirs, the reservoir operator should implement optimization of operational strategies to minimize sediment accumulation in the reservoir area. If necessary, the operators should also take engineering measures such as mechanical desilting to restore reservoir capacity.

Newly cited references (highlighted in red in the manuscript):

Ministry of Water Resources of the People's Republic of China: Code for Reservoir Hydrologic and Sediment Survey, SL339-2006, 2006.

**Minor Comments:**

Comment 1. Lines 60 – 63; and 69: Please add appropriate references.

**Reply:** Thanks for your comment. We have added the corresponding references in Line 62 and 69.

Newly cited references (highlighted in red in the manuscript):

Cao, W. and Liu, C.: Advance and prospect in research on reservoir sediment control and functional restoration, Water Resources and Hydropower Engineering, 49, 1079–1086, https://doi.org/10.13243/j.cnki.slxb.20180655, 2018.

Li, Q., Yu, M., Lu, G., Cai, T., Bai, X., and Xia, Z.: Impacts of the Gezhouba and Three Gorges reservoirs on the sediment regime in the Yangtze River, China, Journal of Hydrology, 403, 224–233, https://doi.org/10.1016/j.jhydrol.2011.03.043, 2011.

Comment 2. Lines 102 – 105: Consider rewriting these sentences for clarity, as the meaning feels vague based on the previous paragraph. In the earlier paragraph, the authors mention that the first method involves topographic surveys to estimate reservoir storage capacity. However, these lines suggest that most reservoirs have not yet conducted storage capacity estimations. Please clarify — do you mean that most methods have not directly estimated storage capacity?

**Reply:** Thanks for your comment. In the previous paragraph, the first method of estimating reservoir capacity by topographic survey was introduced, and then its shortcomings, i.e., long survey period, complex topographic conditions and high cost, were mentioned immediately afterward. Therefore, it is expressed that due to the shortcomings of the methods mentioned in the previous paragraph, most of the reservoirs have not yet conducted storage capacity rechecking, rather than that most methods have not directly used to estimated storage capacity. Lines 103-106 have been rewritten to avoid ambiguity.

Comment 3. Lines 114 - 116: Please add appropriate references.

**Reply:** Thanks for your comment. We have added the corresponding references. Please see Line 124 in the revised manuscript.

Newly cited references (highlighted in red in the manuscript):

Jia, B., Zhou, J., Chen, X., Tian, M., and Zhang, Y.: Fitting reservoir stage-capacity curves and its application in reservoir operation, Journal of Hydroelectric Engineering, 40, 89–99, https://doi.org/DOI: 10.11660/slfdxb.20210209, 2021.

Comment 4. Lines 116 - 118: The logical flow is not smooth; no prior background about flood regulation has been introduced. Please consider revising for better connection and clarity

**Reply:** Thanks for your comment. We have reorganized the structure of the paragraph to ensure articulation and logical clarity by introducing background at the beginning of the paragraph through the impact of sediment on reservoir flood regulation. Please see Lines 107-129 in the revised manuscript.

Comment 5. Line 133: What is meant by "flood prevention operation"? Is this intended to mean "flood control operation"? Please clarify.

**Reply:** Thanks for your detailed comment. We have changed "flood prevention operation" to "flood control operation" and have proofread the entire manuscript. Please see Line 144 in the revised manuscript.

Comment 6. Line 151: A period is missing after "Fig. 1."

**Reply:** Thanks for your detailed comment. We have added the period after "Figure 1." and proofread the entire manuscript. Please see Line 162 in the revised manuscript.

Comment 7. Line 171 and Figure 1: Should "GLT" be corrected to "GLQ"? Please double-check.

**Reply:** Thanks for your detailed comment. We have proofread and standardized the abbreviations throughout the manuscript and added a table of abbreviations. Please see Line 182 in the revised manuscript.

Comment 8. Line 223: Again, a supporting reference is needed here.

**Reply:** Thanks for your comment. We have added the relevant literature for Section 3.2. Please see Lines 242- 246 in the revised manuscript.

Newly cited references (highlighted in red in the manuscript):

Cao, W. and Liu, C.: Advance and prospect in research on reservoir sediment control and functional restoration, Water Resources and Hydropower Engineering, 49, 1079–1086, https://doi.org/10.13243/j.cnki.slxb.20180655, 2018.

Wang, X.: The calculation method of reservoir water level ~ capacity curve based on DEM used by VBA, Water Sciences and Engineering Technology, 31–33, https://doi.org/DOI:10.19733/j.cnki.1672-9900.2018.02.010, 2018.

Comment 9. Lines 239 – 240: Please ensure the formatting of symbols is consistent with the rest of the manuscript.

**Reply:** Thanks for your detailed comment. We have proofread the formatting of formulas and symbols throughout the manuscript.

Comment 10. Line 270: Please specify a unit for "Vrain."

**Reply:** Thanks for your detailed comment. We have added the unit m³ for "$V_{rain}$". Please see Line 291 in the revised manuscript.

Comment 11. The symbol "$I$" is used with different meanings across equations. Please consider using distinct symbols to avoid confusion and ensure consistency.

**Reply:** Thanks for your detailed comment. We have modified the formula variables throughout the manuscript to ensure that there is no mixing of the symbol "$i$".

Comment 12. Line 311: "Figure. 3" should be corrected to "Fig. 3." Please ensure consistent formatting of figure references throughout the manuscript.

**Reply:** Thanks for your detailed comment. We have proofread and standardized the abbreviations throughout the manuscript.

Comment 13. line 411: "Section 3.2" should be "Section 4.2"?

**Reply:** Thanks for your detailed comment. We have corrected "Section 3.2" to "Section 4.2" and proofread the entire manuscript. Please see Line 447 in the revised manuscript.

Comment 14. Line 487: Please add a space before "On the other..."

**Reply:** Thanks for your detailed comment. We have added the missing spaces and proofread the entire manuscript. Please see Line 538 in the revised manuscript.

Comment 15. Lines 507 – 509: Please double-check these sentences for accuracy and consider rewriting them for clarity.

**Reply:** Thanks for your detailed comment. We have reorganized the sentence structure to ensure smooth logic. Please see Lines 578-580 of the revised manuscript.

---

## Author Response (AR1)

**Reviewer #1:**

The study proposes a capacity loss rate (LR) index to reconstruct the water level-storage (WLS) relationship of reservoirs, utilizing measured water/sediment data and operational records. The suitability of the proposed method is evaluated through water balance principles and comparison with traditional approaches. Subsequently, the impact of reconstructed WLS curves on reservoir flood control and operation is quantified using a flood regulation algorithm with varying design inflow scenarios. The experimental design from WLS curve reconstruction to flood operation impact assessment is systematic. While the proposed methods offer practical values for reservoir management, issues remain in language expression and method application. Therefore, I recommend Major Revision for this manuscript. The detailed comments are as follows.

**Response:** Thank you very much for your time in reviewing our manuscript and providing us with invaluable comments, which have substantially improved our manuscript. We appreciated your positive comments on our manuscript. We have carefully considered and addressed all your comments and revised our manuscript accordingly. Please find our detailed point-by-point response to each of your comments below. To facilitate the re-review, we put the comments in **Bold and Black text** and our responses in **Blue plain text**. Additionally, we provided the revised manuscript with changes highlighted in **Red text** to enable you to easily check our revisions. Please note that the line numbers in our responses below refer to those in the revised manuscript, while the line numbers in the comments refer to those in the original manuscript. We hope that our responses and revisions are satisfactory. Thank you very much again for your efforts in re-reviewing our revised manuscript.

Comment 1. The terms "Fig." and "Figure" are used interchangeably in the manuscript. Please standardize to one format for consistency. Additionally, proofreading by a native English speaker should be conducted to improve overall language quality.

**Response:** Thanks for your valuable comment regarding the language in the manuscript. We have standardized all references to "Figure" (e.g., Figure 1) in the revised manuscript. Furthermore, we have engaged a native English speaker to polish our revised manuscript, improving both language and organization.

Comment 2. This study uses the sediment deposition rate (Te) to calculate LR. It is recommended to introduce the concept and development history of Te in Introduction Section and reduce the examples of traditional methods to better highlight the study's core methodology.

**Response:** Thanks for your valuable comment. We have added a brief introduction to the *Te* in the Introduction Section. Defined as the instantaneous ratio of the intercepted sediment to the total sediment load, the trap efficiency has been a key sedimentation

parameter since its conceptualization by Brown (1944). The trap efficiency was extensively investigated using multiple approaches to incorporate the reservoir capacity, watershed characteristics, and sediment load data. Then, the trap efficiency yields distinct estimation methods including the Brown, Brune, and Gill methods (e.g., Brown, 1944; Moragoda et al., 2023; Ren et al., 2024, ...etc.). However, the validity of reconstructing WLSV curves based on sediment accumulation remains understudied (Jia et al., 2021). Meanwhile, reconstruction of the WLSV curve using storage capacity loss estimates induced by sediment accumulation provides a crucial supplement for the traditional reconstruction method (Huang et al., 2018). Please see Lines 114–128 in the revised manuscript.

Newly cited references (highlighted in Red text in the revised manuscript):

Brown, C. B.: Discussion of sedimentation in reservoirs by B. J. Witzig, Proceedings of the American Society of Civil Engineers, 69, 1493–1500, 1944.

Moragoda, N., Cohen, S., Gardner, J., Muñoz, D., Narayanan, A., Moftakhari, H., and Pavelsky, T. M.: Modeling and Analysis of Sediment Trapping Efficiency of Large Dams Using Remote Sensing, Water Resources Research, 59, e2022WR033296, https://doi.org/10.1029/2022WR033296, 2023.

Ren, S., Gao, Y., Wang, W., Zhou, Y., and Zhao, H.: Estimating Sediment Trap Efficiency of Flood Events During Flood Season in the Three Gorges Reservoir, Water Resources Research, 60, e2023WR036975, https://doi.org/10.1029/2023WR036975, 2024.

Tan, G., Chen, P., Deng, J., Xu, Q., Tang, R., Feng, Z., and Yi, R.: Review and improvement of conventional models for reservoir sediment trapping efficiency, Heliyon, 5, e02458, https://doi.org/10.1016/j.heliyon.2019.e02458, 2019.

Comment 3. For 2.1 Section of the study area, in Line 151, a period is missing after "Fig. 1". Please revise throughout the manuscript. In Line 152, the phrase "the Wujiang River Basin has aggravated the interception of sediment" should be reworded to "sediment interception in the basin has increased".

**Response:** Thanks for your valuable comment. We have added the missing period after "Fig. 1" (now written as Figure 1 to be consistent) and reconstructed the sentence to "After the gradual completion of the cascade system in the basin, more sediment has accumulated.". Please see the details in Lines 162–163 in the revised manuscript.

Comment 4. In Methods Section, since the Brune model is an empirical model based on U.S. reservoir siltation survey data, its applicability to reservoir siltation calculation in China may be limited. It is suggested to add a discussion on the model's applicability, supported by relevant literature.

**Response:** Thanks for this valuable comment and the following Comment–6. We have introduced four other empirical models commonly used to compare them with the selected Brune model, in the Section 3.1 (Lines 220–232). The results showed that the Brune model demonstrates better applicability while the Gill and Jothiprakash (coarse sediments) models yielded clearly unreasonable results (notably Te = 1.001 > 1 at the HJD reservoir), the Brown model overestimated Te in three large reservoirs (HJD, WJD, GPT; 0.946–0.990), and the Jothiprakash (medium sediments) model underestimated Te in three small reservoirs (SFY, ST, GLQ; 0.604–0.736). Furthermore, the Te value derived from the Brune model for WJD Reservoir (0.885) closely matched the results reported by Li and Jin (2014) using hydrological station data (0.88). Fu and He (2007) and Tan et al. (2019) additionally confirmed the Brune model's suitability for reservoirs in the Yangtze and Lancang River basins. Please see the details in Lines 529–543 in the revised manuscript.

Newly cited references (highlighted in Red text in the revised manuscript):

Fu, K. and He, D.: Analysis and prediction of sediment trapping efficiencies of the reservoirs in the mainstream of the Lancang River, Chin. Sci. Bull., 52, 134–140, https://doi.org/10.1007/s11434-007-7026-0, 2007.

Li, J. and Jin, Z.: Studying the reservoir sedimentation problem in upper Yangtze River by trap efficiency method, 416–420, 2014.

Tan, G., Chen, P., Deng, J., Xu, Q., Tang, R., Feng, Z., and Yi, R.: Review and improvement of conventional models for reservoir sediment trapping efficiency, Heliyon, 5, e02458, https://doi.org/10.1016/j.heliyon.2019.e02458, 2019.

Comment 5. In 3.1, according to Eq. 1, as  $n_i$  in the denominator increases, the reservoir capacity loss rate LRi also increases. This implies that LR grows over time, suggesting an unrealistic scenario where reservoirs would silt up indefinitely. Please clarify the constraints of this equation to reflect real-world limitations.

**Response:** Thank you for your valuable comment. For physical constraints: According to the study by Hua (2014), the measured sediment data of the WJD Reservoir from 1979 to 2012 showed a capacity loss of 186.8 million m³, which corresponded to LR of 8.12%. Our calculated result (180.3 million m³, LR=7.84%) closely aligns with this value. Further analysis reveals that sediment accumulation in the WJD Reservoir significantly decreased in recent years due to enhanced sediment retention by upstream reservoirs and the "Returning Farmland to Forests and Grasslands" policy. Sediment may also be washed away because the reservoir water level highly varies. These real-world factors collectively constrain the continuous increase in LR.

For mathematical parameter constraints: In Eq. (1),  $n_i$  in the denominator, representing the temporal interval of recorded data, is bounded by the service life of reservoirs (typically $\leq$ 100 years), and imposes a theoretical upper limit on LR. Parameters  $Q_i$  (multi-year average inflow water volume at the dam site) and  $w_i$  (long-term average

sediment in the inflow water) are dynamic variables. Their values evolve with watershed management practices, reservoir operation strategies, and climatic changes, which further curb the long-term growth trend of LR.

In summary, Eq. (1) inherently incorporates constraints from both physical processes and mathematical parameters to ensure that LR cannot indefinitely increase when  $n_i$  increases. We have added more descriptions and clarifications on this matter in the Lines 225–227 of the revised manuscript.

Comment 6. The study establishes Brune model to calculate Te, but multiple approaches exist for this purpose. It is recommended to compare the results with those from other models in the Methods section to strengthen methodological robustness.

**Response:** Thanks for your valuable comment. We followed your suggestion to test and compare more methods. We have introduced four other empirical models commonly used to compare them with the selected Brune model, in Section 3.1 (Lines 220–232). The results showed that the Brune model demonstrates better applicability, while the Gill and Jothiprakash (coarse sediments) models yielded clearly unreasonable results (notably Te = 1.001 > 1 at the HJD reservoir), the Brown model overestimated Te in three large reservoirs (HJD, WJD, GPT; 0.946–0.990), and the Jothiprakash (medium sediments) model underestimated Te in three small reservoirs (SFY, ST, GLQ; 0.604–0.736). Please see Lines 342–358 in the revised manuscript.

Comment 7. The reservoir exhibits complicated morphometry, with significant variations in water depth and width. In the current study, several reservoirs (e.g., Hongjiadu and Goupitan) also cover extensive surface areas. On the other hand, the authors employ mathematical functions with exponential, polynomial and power function forms in Eqs. (3-5) to estimate reservoir water storage changes. Please address potential errors induced by the simplified estimation formula. Could the accuracy of the WLS curve be improved by subdividing the reservoir into smaller sections based on morphometry and summing their contributions?

**Response:** Thanks for your constructive comment. Although summing several subsections could improve accuracy, it would require detailed bathymetric data (e.g., high–resolution DEMs of reservoir areas, measured underwater topography, and cross-sectional hydrological profiles). Unfortunately, these detailed bathymetric data are unavailable for most reservoirs, and the same is true for our present study. We have used various mathematical function types including exponential, polynomial and power functions in the quantitative representation of WLSV curve to reduce modeling errors. Additionally, we plan to explore more refined methods in future studies.

Comment 8. In Eqs. (3-5), the water level (Z) is treated as the response variable, while the reservoir water storage is the explanatory variable. Yet, the reservoir WLS curve reconstruction is evaluated based on the water storage variable. This inconsistency needs clarification.

**Response:** Thanks for your valuable comment. In Eqs. (2)–(4), when water level (Z) is treated as the explanatory variable, the function exhibits relatively poor fitting performance for the WLS curve. For instance, the R2 is 0.91 at HJD Reservoir. However, the R2 improves to 0.99 when water level (Z) is instead designated as the response variable. Consequently, we selected water level (Z) as the response variable in our study. For practical applications, the mathematical function can be inverted to derive water level (Z) as the explanatory variable. Please see more necessary clarifications in Lines 242–246 in the revised manuscript.

**Comment 9. In Figure 1, the abbreviation for the "GLT" reservoir should be corrected to "GLQ" (Geliqiao).**

**Response:** Thanks for your comment. The abbreviation for "Geliqiao Reservoir" in Figure 1 has been corrected to "GLQ" throughout the manuscript. Please see Line 180 in the revised manuscript.

Comment 10. The Discussion Section should include a comparison of your WLS curve construction method with other existing approaches. Additionally, please elaborate on how the accuracy of WLS curve impacts the reliability of the water storage estimation.

**Response:** Thank you for your valuable comment. Direct accuracy assessment through a WLSV curve reconstructed by traditional field measurements with surveying and mapping technology is undoubtedly the optimal validation method. However, the implementation of the traditional method is often limited by high costs, complex topography, and lengthy time requirements. Consequently, up—to—date and accurate insitu WLSV curves for the studied reservoirs are unavailable for most reservoirs around the globe. Because there are no direct validation data, we cannot directly quantify the accuracy of the reconstructed WLSV curves in this study and must develop the alternative and indirect reconstruction approach, which is the core motivation for our study. Following your suggestions, we have supplemented the comparison of our reconstruction method for the WLSV curve with other approaches. Gui et al. (2025) reconstructed the WLSV curves of the HJD Reservoir using Sentinel—1 SAR data and estimated capacity losses of 65 million m³. Our estimated sediment-induced capacity loss for the same reservoir (69.5 million m³) is strongly consistent with these results.

Furthermore, we evaluated the applicability of five classic *Te* empirical models in the Wujiang River Basin. The results indicated that the Gill and Jothiprakash models (coarse sediments) performed poorly and yielded an unrealistic *Te* of 1.001 at the HJD Reservoir. At the WJD Reservoir, the Brown model and Jothiprakash model (medium sediments) overestimated *Te* (0.946 and 0.932, respectively) compared with the reference value of 0.88 from sediment measurements (Li and Jin, 2014). By contrast, the Brune model provided a *Te* estimate of 0.885, which closely matched the observed data. This validation confirms the accuracy of the Brune model for *Te* estimation in the Wujiang River Basin, which is consistent with its established applicability in the

Yangtze and Lancang basins (Fu and He, 2007; Tan et al., 2019). Furthermore, the spatial pattern of storage capacity loss volumes and rates is reasonable among different reservoirs in the cascade. Both larger capacity loss volumes and rates occur in the four reservoirs at the middle and upper reaches of the river. This result is highly consistent with the conclusion of more severe sediment accumulation in the upper reaches and the priority of the upper reservoirs to trap sands (Brush, 1989; Yuan et al., 2022). This fundamental pattern indirectly offers further evidence or the reasonableness for the reconstructed WLSV curves.

In addition, we supplemented the discussion on the potential effects of uncertainties in WLSV curves on the estimation accuracy of water level and operational reliability. This concern is substantiated by a previous study on the Three Gorges Reservoir (Jia et al., 2021), which demonstrated that the reasonableness and uncertainty of WLSV curves critically impact operational reliability. Specifically, the maximum errors in simulated water level and hydropower output were 3.0 m and  $50 \times 10^4$  kW when raw WLSV scatter points were used but significantly decreased to 2.2 m and  $29 \times 10^4$  kW when a fitted WLSV curve was used. These results clearly show that inherent uncertainties in WLSV curves can cause substantial deviations in reservoir scheduling calculations. Consequently, such uncertainties directly impact critical operational aspects, including power generation efficiency and flood risk management.

We stress the need for future studies that conduct the in-situ bathymetric survey and address the issue of direct validation for the reasonableness of the reconstructed WLSV curve. We have added more discussion in Lines 507–519 and 529–543 in the revised manuscript.

Newly cited references (highlighted in Red text in the revised manuscript):

- Gui, X., Ma, Q., Li, J., Duan, Z., Xiong, L., and Xu, C.-Y.: Reconstructing Reservoir Water Level-Area-Storage Volume Curve Using Multi-source Satellite Imagery and Intelligent Classification Algorithms, Water Resour Manage, https://doi.org/10.1007/s11269-025-04205-7, 2025.
- Li, J. and Jin, Z.: Studying the reservoir sedimentation problem in upper Yangtze River by trap efficiency method, 416–420, 2014.
- Fu, K. and He, D.: Analysis and prediction of sediment trapping efficiencies of the reservoirs in the mainstream of the Lancang River, Chin. Sci. Bull., 52, 134–140, https://doi.org/10.1007/s11434-007-7026-0, 2007.
- Tan, G., Chen, P., Deng, J., Xu, Q., Tang, R., Feng, Z., and Yi, R.: Review and improvement of conventional models for reservoir sediment trapping efficiency, Heliyon, 5, e02458, https://doi.org/10.1016/j.heliyon.2019.e02458, 2019.

Brush, G. S.: Rates and patterns of estuarine sediment accumulation, Limnology & Oceanography, 34, 1235–1246, https://doi.org/10.4319/lo.1989.34.7.1235, 1989.

Yuan J., Chen W., Yang C., and Xiong M.: Study on Sediment Retention and Reduction of Reservoirs in Wujiang River Basin, JWRR, 11, 249–259, https://doi.org/10.12677/JWRR.2022.113027, 2022.

Jia, B., Zhou, J., Chen, X., Tian, M., and Zhang, Y.: Fitting reservoir stage-capacity curves and its application in reservoir operation, Journal of Hydroelectric Engineering, 40, 89–99, https://doi.org/DOI: 10.11660/slfdxb.20210209, 2021.

Comment 11. The abbreviations in the text and figures should be independent of each other. It is recommended that the corresponding vocabulary of the reservoir name abbreviations should also be clearly stated in the data methods in the manuscript text.

**Response:** Thanks for the useful comment. A comprehensive list of abbreviations has been added, explicitly defining the abbreviation conventions for all reservoirs' names. The entire manuscript has been revised to consistently use abbreviations (with full names introduced at first mention) in both text and figure.

**Table 1.** Full names used in articles and their corresponding abbreviations

| Full name                  | Abbreviation  |
|----------------------------|---------------|
| water level-storage volume | WLSV          |
| loss rate                  | LR            |
| Hongjiadu                  | HJD           |
| Dongfeng                   | DF            |
| Suofengying                | SFY           |
| Wujaingdu                  | WJD           |
| Goupitan                   | GPT           |
| Silin                      | $\mathbf{SL}$ |
| Shatuo                     | ST            |
| Dahuashui                  | DHS           |
| Geliqiao                   | GLQ           |

We would like to thank Reviewer#1 again for your time and efforts in re-reviewing our revised manuscript. We hope our responses and revisions are satisfactory.

**Reviewer #2:**

This paper proposes a method to reconstruct the water level-storage (WLS) relationship of reservoirs by estimating storage capacity loss due to sediment accumulation. The method is applied to a cascade of nine reservoirs along China's Wujiang River, and the reconstructed WLS curves are validated using water balance analysis and DEM-based surface data. Overall, the framework presented in this study provides useful insights and offers practical value for reservoir management to a certain extent. However, the manuscript would benefit from further refinement to improve its clarity, logical flow, and rigor before it can be considered for publication. In particular, some sections of the paper would benefit from a smoother transition between sentences and a clearer presentation of the methodology and results. Below, I outline specific comments and suggestions for the authors' consideration.

**Reply:** Thank you very much for your time in reviewing our manuscript and providing us with invaluable comments, which have substantially improved our manuscript. We appreciated your positive comments on our manuscript. We have carefully considered and addressed all your comments and revised our manuscript accordingly. Please find our detailed point-by-point response to each of your comments below. To facilitate the re-review, we put the comments in **Bold and Black text** and our responses in **Blue plain text**. Additionally, we provided the revised manuscript with changes highlighted in **Red text** to enable you to easily check our revisions. Please note that the line numbers in our responses below refer to those in the revised manuscript, while the line numbers in the comments refer to those in the original manuscript. We hope that our responses and revisions are satisfactory. Thank you very much again for your efforts in re-reviewing our revised manuscript.

Comment 1. I recommend that the authors consider adjusting or restructuring the paragraphs in the introduction section. The logical flow and overall storyline of the Introduction are currently not entirely smooth. For example, the authors state that "Estimating the storage capacity loss induced by sediment accumulation and reconstructing the WLS curve provides a new direction for the second category of methods," but no clear background or justification is provided to connect this statement with the earlier discussion. Prior sentences only mention that sediment accumulation leads to reservoir storage loss, without adequately building the context for why reconstructing the WLS curve becomes a new methodological direction. Additional examples of logical inconsistencies are noted in my minor comments section.

**Response:** Thanks for your valuable comment. We have restructured the paragraph of the Introduction Section in Lines 105–128 in the revised manuscript. To give you an overview of our revisions for this part, below summarizes what we have revised.

The possible effects of sediment on reservoirs, especially on flood control operation, are introduced first. Then, the development of *Te* is described to better highlight the core methodology. Finally, the shortcomings investigated by current studies are summarized, for example, few studies have investigated the validity of reconstructing the WLSV curve based on sediment accumulation, and there has been insufficient attention to the practical effectiveness of the reconstructed curve. Therefore, Using the estimation of storage capacity loss induced by sediment accumulation to reconstruct the WLSV curve provides a new direction for the second category of methods for reconstructing WLSV curves.

Newly cited references (highlighted in Red text in the revised manuscript):

Brown, C. B.: Discussion of sedimentation in reservoirs by B. J. Witzig, Proceedings of the American Society of Civil Engineers, 69, 1493–1500, 1944.

Moragoda, N., Cohen, S., Gardner, J., Muñoz, D., Narayanan, A., Moftakhari, H., and Pavelsky, T. M.: Modeling and Analysis of Sediment Trapping Efficiency of Large Dams Using Remote Sensing, Water Resources Research, 59, e2022WR033296, https://doi.org/10.1029/2022WR033296, 2023.

Ren, S., Gao, Y., Wang, W., Zhou, Y., and Zhao, H.: Estimating Sediment Trap Efficiency of Flood Events During Flood Season in the Three Gorges Reservoir, Water Resources Research, 60, e2023WR036975, https://doi.org/10.1029/2023WR036975, 2024.

Tan, G., Chen, P., Deng, J., Xu, Q., Tang, R., Feng, Z., and Yi, R.: Review and improvement of conventional models for reservoir sediment trapping efficiency, Heliyon, 5, e02458, https://doi.org/10.1016/j.heliyon.2019.e02458, 2019.

Comment 2. I suggest that the authors add appropriate references throughout the manuscript, particularly when providing background information in the Study Area and Data section (e.g., indicating the sources of the data) and in the Methodology section. Providing references would enhance the transparency and credibility of the study. In addition, it would be helpful if the authors could clarify the rationale behind the selection of the study area and the specific reservoirs.

**Response:** Thanks for your valuable comment. We have added the corresponding references (Wu et al., 2018; Yuan et al., 2022) to the Study Area Section when introducing the background of the Wujiang River Basin. In the Data Section, the water and sediment data utilized in this study were obtained from a professional engineering organization under a data-sharing agreement. We conducted rigorous quality control for the data, involving manual verification and rectification of anomalous and erroneous data. Please see details in Lines 148–189 in the revised manuscript.

Newly cited references (highlighted in Red text in the revised manuscript):

Wu, X., Xiang, X., Chen, X., Zhang, X., and Hua, W.: Effects of cascade reservoir dams on the streamflow and sediment transport in the Wujiang River basin of the Yangtze River, China, Inland Waters, 8, 216–228, https://doi.org/10.1080/20442041.2018.1457850, 2018.

Yuan J., Chen W., Yang C., and Xiong M.: Study on Sediment Retention and Reduction of Reservoirs in Wujiang River Basin, JWRR, 11, 249–259, https://doi.org/10.12677/JWRR.2022.113027, 2022.

Comment 3. For Equations (1) and (2), the authors make certain assumptions to simplify the derivation given that the loss rate (LR) is a key indicator in this manuscript, I suggest that the authors provide more detailed explanations of these assumptions for the benefit of the readers. In particular, it would be helpful to clarify why these assumptions were made, how they affect the methodology, and whether there are existing references that support or justify them.

**Response:** Thank you for your valuable comment. This study used the multi-year average sediment concentration data of inflow water for each reservoir. The calculated results based on these data reflect the multi-year average reservoir capacity loss; thus, we assumed a uniform sediment distribution at the bottom and a constant annual deposition velocity. If detailed annual sediment concentration data of reservoir inflows were available, the proposed method could be applied to estimate the annual reservoir capacity loss. Following your suggestion, we have added more descriptions and clarification in Lines 210–214 in the revised manuscript.

Comment 4. I suggest that the authors consider creating a table listing all abbreviations used throughout the manuscript for easier reference. Additionally, while the abbreviations for each reservoir are defined and used in the figures and text, the writing in Section 4.1 inconsistently switches between full names and abbreviations. For consistency and to improve readability, I recommend using the reservoir abbreviations throughout the text, especially since the figures present the abbreviations as well.

**Response:** Thanks for your comment. A comprehensive list of abbreviations has been added, explicitly defining the abbreviation conventions for reservoir names. The entire manuscript has been revised to consistently use the same abbreviations (with full names introduced at first occurrence) in both text and figure.

**Table 1.** Full names used in articles and their corresponding abbreviations

| Full name                  | Abbreviation |
|----------------------------|--------------|
| water level-storage volume | WLSV         |
| loss rate                  | LR           |
| Hongjiadu                  | HJD          |
| Dongfeng                   | DF           |
| Suofengying                | SFY          |
| Wujaingdu                  | WJD          |
| Goupitan                   | GPT          |
| Silin                      | SL           |
| Shatuo                     | ST           |
| Dahuashui                  | DHS          |
| Geliqiao                   | GLQ          |

Comment 5. The discussion section should be strengthened and expanded. Specifically, the authors could provide a more detailed comparison of their WLS curve reconstruction method with other existing approaches. Additionally, the current discussion (Lines 491–499) may not be sufficient to fully demonstrate the reasonableness of the reconstructed WLS curves. I encourage the authors to elaborate further on the topic of "Reasonableness and uncertainty in reconstructed reservoir WLS curves".

Response: Thank you for your valuable comment. Direct accuracy assessment through a WLSV curve reconstructed by traditional field measurements with surveying and mapping technology is undoubtedly the optimal validation method. However, the implementation of the traditional method is often limited by high costs, complex topography, and lengthy time requirements. Consequently, up—to—date and accurate insitu WLSV curves for the studied reservoirs are unavailable for most reservoirs around the globe. Because there are no direct validation data, we cannot directly quantify the accuracy of the reconstructed WLSV curves in this study and must develop the alternative and indirect reconstruction approach, which is the core motivation for our study. Following your suggestions and the other reviewer's, we have supplemented the comparison of our reconstruction method for the WLSV curve with other approaches. Gui et al. (2025) reconstructed the WLSV curves of the HJD Reservoir using Sentinel—1 SAR data and estimated capacity losses of 65 million m³. Our estimated sediment—induced capacity loss for the same reservoir (69.5 million m³) is strongly consistent with these results.

Furthermore, we evaluated the applicability of five classic *Te* empirical models in the Wujiang River Basin. The results indicated that the Gill and Jothiprakash models (coarse sediments) performed poorly and yielded an unrealistic *Te* of 1.001 at the HJD Reservoir. At the WJD Reservoir, the Brown model and Jothiprakash model (medium sediments) overestimated *Te* (0.946 and 0.932, respectively) compared with the

reference value of 0.88 from sediment measurements (Li and Jin, 2014). By contrast, the Brune model provided a *Te* estimate of 0.885, which closely matched the observed data. This validation confirms the accuracy of the Brune model for *Te* estimation in the Wujiang River Basin, which is consistent with its established applicability in the Yangtze and Lancang basins (Fu and He, 2007; Tan et al., 2019). Furthermore, the spatial pattern of storage capacity loss volumes and rates is reasonable among different reservoirs in the cascade. Both larger capacity loss volumes and rates occur in the four reservoirs at the middle and upper reaches of the river. This result is highly consistent with the conclusion of more severe sediment accumulation in the upper reaches and the priority of the upper reservoirs to trap sands (Brush, 1989; Yuan et al., 2022). This fundamental pattern indirectly offers further evidence or the reasonableness for the reconstructed WLSV curves.

We stress the need for future studies that conduct the in-situ bathymetric survey and address the issue of direct validation for the reasonableness of the reconstructed WLSV curve. We have added more discussion in Lines 507–519 and 529–543 in the revised manuscript.

Newly cited references (highlighted in Red text in the revised manuscript):

- Gui, X., Ma, Q., Li, J., Duan, Z., Xiong, L., and Xu, C.-Y.: Reconstructing Reservoir Water Level-Area-Storage Volume Curve Using Multi-source Satellite Imagery and Intelligent Classification Algorithms, Water Resour Manage, https://doi.org/10.1007/s11269-025-04205-7, 2025.
- Li, J. and Jin, Z.: Studying the reservoir sedimentation problem in upper Yangtze River by trap efficiency method, 416–420, 2014.
- Fu, K. and He, D.: Analysis and prediction of sediment trapping efficiencies of the reservoirs in the mainstream of the Lancang River, Chin. Sci. Bull., 52, 134–140, https://doi.org/10.1007/s11434-007-7026-0, 2007.
- Tan, G., Chen, P., Deng, J., Xu, Q., Tang, R., Feng, Z., and Yi, R.: Review and improvement of conventional models for reservoir sediment trapping efficiency, Heliyon, 5, e02458, https://doi.org/10.1016/j.heliyon.2019.e02458, 2019.
- Brush, G. S.: Rates and patterns of estuarine sediment accumulation, Limnology & Oceanography, 34, 1235–1246, https://doi.org/10.4319/lo.1989.34.7.1235, 1989.
- Yuan J., Chen W., Yang C., and Xiong M.: Study on Sediment Retention and Reduction of Reservoirs in Wujiang River Basin, JWRR, 11, 249–259, https://doi.org/10.12677/JWRR.2022.113027, 2022.

Comment 6. Could the authors elaborate on the potential policy implications of this study? How can decision-makers apply these findings to real-world water resource management?

**Response:** Thanks for your valuable comment. In the newly added Section 5.2 (Lines 554–572), we analyzed the policy implications arising from the study's findings and their insights for reservoir management, as follows: As current design WLSV curves underestimate flood regulation risk, reservoir operations should strengthen sediment accumulation monitoring and conduct periodic recalibration of WLSV curves for high LR reservoirs as mandated by Code for Reservoir Hydrologic and Sediment Survey (2006), to mitigate the negative impacts on the operational efficiency. Furthermore, reservoirs in the upper-middle Wujiang River suffer serious capacity loss from sediment accumulation. For such reservoirs, the reservoir operator should increase sediment discharge by using the propagation time difference between flood peaks and sediment peaks to modulate the reservoir discharge hydrograph during flood season, while ensuring flood control safety (Ren et al., 2021). Simultaneously, sediment reduction operations should be implemented by optimizing the reservoir water level drawdown process in coordination with incoming water and sediment conditions, thereby promoting scouring of sediment accumulation (Wang et al., 2016). The reservoirs that storage loss is severe and the sediment accumulation is relatively coarse (making scouring difficult), comprehensive measures such as mechanical dredging and engineering remediation should be promptly implemented to partially recover the effective storage capacity lost to sediment accumulation.

Newly cited references (highlighted in Red text in the revised manuscript):

Ministry of Water Resources of the People's Republic of China: Code for Reservoir Hydrologic and Sediment Survey, SL339-2006, 2006.

Ren, S., Zhang, B., Wang, W.-J., Yuan, Y., and Guo, C.: Sedimentation and its response to management strategies of the Three Gorges Reservoir, Yangtze River, China, CATENA, 199, 105096, https://doi.org/10.1016/j.catena.2020.105096, 2021.

Wang, B., Yan, D., Wen, A., and Chen, J.: Influencing factors of sediment deposition and their spatial variability in riparian zone of the Three Gorges Reservoir, China, J. Mt. Sci., 13, 1387–1396, https://doi.org/10.1007/s11629-015-3806-1, 2016.

**Minor Comments:**

Comment 1. Lines 60 – 63; and 69: Please add appropriate references.

**Response:** Thanks for your comment. We have added the corresponding references in Lines 61 and 67.

Newly cited references (highlighted in Red text in the revised manuscript):

Cao, W. and Liu, C.: Advance and prospect in research on reservoir sediment control and functional restoration, Water Resources and Hydropower Engineering, 49, 1079–1086, https://doi.org/10.13243/j.cnki.slxb.20180655, 2018.

Li, Q., Yu, M., Lu, G., Cai, T., Bai, X., and Xia, Z.: Impacts of the Gezhouba and Three Gorges reservoirs on the sediment regime in the Yangtze River, China, Journal of Hydrology, 403, 224–233, https://doi.org/10.1016/j.jhydrol.2011.03.043, 2011.

Comment 2. Lines 102 – 105: Consider rewriting these sentences for clarity, as the meaning feels vague based on the previous paragraph. In the earlier paragraph, the authors mention that the first method involves topographic surveys to estimate reservoir storage capacity. However, these lines suggest that most reservoirs have not yet conducted storage capacity estimations. Please clarify — do you mean that most methods have not directly estimated storage capacity?

**Response:** Thanks for your comment. In the previous paragraph, the first method of estimating reservoir capacity by topographic survey was introduced, and then its shortcomings, i.e., long survey period, complex topographic conditions and high cost, were mentioned immediately afterward. Therefore, it is expressed that due to the shortcomings of the methods mentioned in the previous paragraph, most of the reservoirs have not yet conducted storage capacity rechecking, rather than that most methods have not been directly used to estimated storage capacity. Lines 101–104 have been rewritten to avoid the ambiguity.

**Comment 3. Lines 114 - 116: Please add appropriate references.**

**Response:** Thanks for your comment. We have added the corresponding references. Please see Lines 122–123 in the revised manuscript.

Newly cited references (highlighted in Red text in the revised manuscript):

Jia, B., Zhou, J., Chen, X., Tian, M., and Zhang, Y.: Fitting reservoir stage-capacity curves and its application in reservoir operation, Journal of Hydroelectric Engineering, 40, 89–99, https://doi.org/DOI: 10.11660/slfdxb.20210209, 2021.

Comment 4. Lines 116 - 118: The logical flow is not smooth; no prior background about flood regulation has been introduced. Please consider revising for better connection and clarity.

**Response:** Thanks for your comment. We have reorganized the structure of the paragraph to ensure articulation and logical clarity by introducing background at the beginning of the paragraph through the impact of sediment on reservoir flood regulation. Please see Lines 105–128 in the revised manuscript.

Comment 5. Line 133: What is meant by "flood prevention operation"? Is this intended to mean "flood control operation"? Please clarify.

**Response:** Thanks for your detailed comment. We have changed "flood prevention operation" to "flood control operation" and have proofread the entire manuscript. Please see Line 143 in the revised manuscript.

Comment 6. Line 151: A period is missing after "Fig. 1."

**Response:** Thanks for your detailed comment. We have added the period after "Figure 1." and proofread the entire manuscript. Please see Line 162–163 in the revised manuscript.

Comment 7. Line 171 and Figure 1: Should "GLT" be corrected to "GLQ"? Please double-check.

**Response:** Thanks for your detailed comment. We have proofread and standardized the abbreviations throughout the manuscript and added a table of abbreviations. Please see Line 180 in the revised manuscript.

Comment 8. Line 223: Again, a supporting reference is needed here.

**Response:** Thanks for your comment. We have added the relevant literature for Section 3.2. Please see Lines 241 in the revised manuscript.

Newly cited references (highlighted in Red text in the revised manuscript):

Cao, W. and Liu, C.: Advance and prospect in research on reservoir sediment control and functional restoration, Water Resources and Hydropower Engineering, 49, 1079–1086, https://doi.org/10.13243/j.cnki.slxb.20180655, 2018.

Wang, X.: The calculation method of reservoir water level ~ capacity curve based on DEM used by VBA, Water Sciences and Engineering Technology, 31–33, https://doi.org/DOI:10.19733/j.cnki.1672-9900.2018.02.010, 2018.

Comment 9. Lines 239 - 240: Please ensure the formatting of symbols is consistent with the rest of the manuscript.

**Response:** Thanks for your detailed comment. We have proofread the formatting of formulas and symbols throughout the manuscript.

Comment 10. Line 270: Please specify a unit for "Vrain."

**Response:** Thanks for your detailed comment. We have added the unit m3 for " $V_{\text{rain}}$ ". Please see Line 290 in the revised manuscript.

Comment 11. The symbol "P" is used with different meanings across equations. Please consider using distinct symbols to avoid confusion and ensure consistency.

**Response:** Thanks for your detailed comment. We have modified the formula variables throughout the manuscript to ensure that there is no mixing of the symbol "i".

Comment 12. Line 311: "Figure. 3" should be corrected to "Fig. 3." Please ensure consistent formatting of figure references throughout the manuscript.

**Response:** Thanks for your detailed comment. We have proofread and standardized the abbreviations throughout the manuscript.

Comment 13. line 411: "Section 3.2" should be "Section 4.2"?

**Response:** Thanks for your detailed comment. We have corrected "Section 3.2" to "Section 4.2" and proofread the entire manuscript. Please see Line 441 in the revised manuscript.

Comment 14. Line 487: Please add a space before "On the other...".

**Response:** Thanks for your detailed comment. We have proofread the entire manuscript to avoid such issues.

Comment 15. Lines 507 - 509: Please double-check these sentences for accuracy and consider rewriting them for clarity.

**Response:** Thanks for your detailed comment. We have reorganized the sentence structure to ensure smooth logic. Please see Lines 579–582 of the revised manuscript.

We would like to thank Reviewer#2 again for your time and efforts in re-reviewing our revised manuscript. We hope our responses and revisions are satisfactory.

---

## Author Response (AR2)

**Dear Editor and Reviewers:**

We sincerely appreciate your efforts in processing our manuscript. In this round of minor revision, we addressed the reviewer comments point-by-point. Additionally, we also re-checked and corrected the grammar, expression, and formatting issues throughout the manuscript.

Below is our point-by-point response to each of your comments. To facilitate the rereview process, we have presented the comments in **Bold and Black text** and our
responses in **Blue plain text**. Additionally, we have provided the revised manuscript
with all changes highlighted in **Red text** for easy reference. Please note that the line
numbers in our responses refer to those in the revised manuscript, whereas the line
numbers in the comments correspond to the original manuscript. Look forward to your
decision.

Best regards,

Qiumei Ma on behalf of all coauthors

**Reviewer #1:**

The revised manuscript has addressed all my questions from the original version. The clarity, logical rigor, and the inter-sentence cohesion within sections have now been significantly improved. While several minor issues remain, I recommend Minor Revision for this version. The detailed comments are as follows.

**Response:** Thank you very much for reviewing our revised manuscript again and for providing us with additional valuable comments, which have been instrumental in further improving our manuscript. We hope that our responses and revisions meet your expectations.

Comment 1. Variables in the manuscript, such as "Z\*" and " $\gamma$ ", should be italicized consistently like other variables.

**Response:** Thanks for your detailed comment. We have reviewed the entire manuscript to ensure that variables such as " $Z^*$ " and " $\gamma$ " are italicized consistently with other variables.

Comment 2. The usage of the hyphens "-" and en dashes "-" is mixed throughout the manuscript and should be standardized.

**Response:** Thanks for your detailed comment. We have checked the entire manuscript to ensure that both hyphens "-" and en dashes "-" are used correctly.

Comment 3. The symbol "Te" is used in Table 3, while "Tei" is used in Table 4. These should be made consistent.

**Response:** Thanks for your detailed comment. All relevant notations in the manuscript have been unified to " $Te_i$ ", for example in Lines 343–354.

Comment 4. In Lines 554 - 572 of Section 5.2, the logical flow could be strengthened. Please consider revising for better connection and clarity.

**Response:** Thanks for your detailed comment. We have revised the transitional words in that paragraph to improve its coherence and logical clarity. We first highlight the general suggestion like strengthen sediment accumulation monitoring and conduct periodic recalibrations of WLSV curves, then moves to specific operational strategies for different sedimentation scenarios—first for reservoirs with severe capacity loss, and second for those where sediment accumulation is coarse and difficult to scour. The revisions better emphasize this logical progression from general policy to tailored measures. Please see Lines 577–574 in the revised manuscript.

**Comment 5. For Lines 161 – 165, please add appropriate references.**

**Response:** Thanks for your detailed comment. We have added the corresponding references. Please see Lines 165 in the revised manuscript.

Newly cited references (highlighted in red in the revised manuscript):

Yuan, J. and Xu, Q.: Sediment trapping effect by reservoirs in the Jinsha River basin, Advances in Water Science, 29, 482–491, https://doi.org/DOI:10.14042/j.cnki.32.1309.2018.04.004, 2018.